# Do Think Tags Really Help LLMs Plan?
# A Critical Evaluation of ReAct-Style Prompting

**Siddhant Bhambri**[*]                                                                 *siddhantbhambri@asu.edu*
*School of Computing and AI*
*Arizona State University*

**Mudit Verma**[*†]                                                                        *muditvema@asu.edu*
*School of Computing and AI*
*Arizona State University*

**Subbarao Kambhampati**                                                            *rao@asu.edu*
*School of Computing and AI*
*Arizona State University*

**Reviewed on OpenReview:** *https://openreview.net/forum?id=aFAMPSmNHR*

## Abstract

The reasoning abilities of Large Language Models (LLMs) remain a topic of considerable interest and debate. Among the original papers arguing for emergent reasoning abilities of LLMs, ReAct became particularly popular by claiming to tease out LLM reasoning abilities with special prompting involving "*interleaving reasoning trace with action execution*". In this paper, we critically examine the claims of ReAct-style prompting for planning and sequential decision-making problems. By introducing systematic variations to the input prompt, we perform a sensitivity analysis along the original claims of ReAct. Our experiments in AlfWorld and WebShop, domains that were used in the original ReAct work, show that the performance is minimally influenced by the interleaved reasoning trace or by the content of these generated reasoning traces. Instead, the performance of LLMs is primarily driven by the unreasonably high degree of similarity between input example tasks and queries, with shockingly little ability to generalize. In addition to raising questions on claims about reasoning abilities, this lack of generalization also implicitly forces the prompt designer to provide instance-specific examples, significantly increasing the cognitive burden on the human. Our empirical results show that the perceived reasoning abilities of LLMs stem from the exemplar-query similarity and approximate retrieval rather than any inherent reasoning abilities, thereby leading to severe lack of generalization beyond the few-shot examples given in the prompts. Our code and prompt settings can be found here on GitHub.

## 1 Introduction

Large Language Models (LLMs) have seen rapid advancements specifically in Natural Language Processing and Understanding (NLP & NLU). LLMs have unparalleled capabilities in text generation, summarization, translation, and question answering to name a few (Bubeck et al., 2023). Motivated by these capabilities of LLMs, there has also been a rush to look for other emergent abilities–especially for reasoning and planning. A popular way to improve LLM performance on reasoning/planning tasks has been in-context prompting or prompt engineering (Sahoo et al., 2024) to include instructions (Giray, 2023), syntax structure (Marvin et al., 2023), criticism and plan guidance with verification (Kambhampati et al., 2024), etc. Among these approaches, ReAct (Yao et al., 2022b), presented at ICLR 2023, stands out for its claims to improve LLM

---

[*]Equal Contribution
[†]Work done while a PhD student at ASU, now at Google DeepMind

planning abilities through the use of reasoning traces interleaved with action execution given as plan guidance. Given the conflicting reports about the planning abilities of LLMs in the literature Valmeekam et al. (2024), in this paper we are particularly interested in evaluating these claims.

In our initial experiments with ReAct for planning, we found that the system is overly dependent on a high degree of syntactic similarity between the example prompt and the query, and is extremely brittle to minor perturbations to the input prompt. For example, when provided with an explicit set of examples of *pick-and-place-object* task and asked to plan for a *pick-and-place-two-objects* task, it should be trivial to generalize the solution of the examples to the queried task. Unfortunately, even such a minor variation to the original ReAct agent setup disrupts its performance.

Given the seemingly widespread adoption of ReAct methodology (as of this writing, it has 2,411 citations), the brittleness we witnessed calls for a systematic study of the factors contributing to the performance of ReAct-based LLM agents. Based on the claims of (Yao et al., 2022b), we isolate three possible reasons for the claimed performance of ReAct framework: 1) the utility of interleaving reasoning trace during action execution, 2) the utility of providing plan guidance, and, 3) the significance of example prompt provided to the LLM.

In this work, we systematically evaluate the brittleness of ReAct by studying which potential factors contribute to its performance. This analysis is conducted by investigating the following research questions- **RQ1:** Does the agent performance depend on interleaving reasoning trace with action execution? **RQ2:** How does the nature of the reasoning trace or guidance information affect the performance of LLM agents? **RQ3:** How does the similarity between the example ⟨problem, solution⟩and the query ⟨problem, ? ⟩, which are present in the prompt, affect LLM agent performance?

We conduct extensive experiments on the AlfWorld and WebShop domain using various LLM models, including GPT-3.5-turbo, GPT-3.5-instruct, GPT-4-0314(Old Variant), GPT-4-0613 (Latest Variant), GPT-4o, Claude-Opus and Llama 3.1-8B. Through our comprehensive empirical study, we answer each of the research questions above. (RQ1) Rather than being dependent on reasoning trace being interleaved with action execution, we find that LLM performance in fact improves when the reasoning trace is **not** interleaved with action execution. (RQ2) Moreover, providing weaker guidance or placebo-guidance (where the text provides no information about the task) has comparable performance to strong reasoning trace-based guidance. Answers to RQ1 and RQ2 suggest that the source of ReAct LLM agent performance is **not** the interleaving aspect or the content of the reasoning trace. Finally, in (RQ3), we see that variations to the example prompt such that it belongs to a very similar task within the same domain, or just has a different goal or plan attributes than the queried problem; causes the performance of ReAct-based LLM agent to plummet, highlighting the lack of length and inter-task generalization.

This lack of generalization leads to significant brittleness with respect to prompt design. Depending on the domain and task, prompt designers may provide abstract guidance, task-specific instructions within the same domain, analogical examples, or global reasoning traces. ReAct's inability to robustly adapt to such variations underscores its limitations in handling diverse input prompts, thereby necessitating a closer examination of its design and implementation.

Our findings highlight that the benefits of ReAct-style prompting are limited to cases where prompt engineers can curate highly instance-specific examples. This may not scale for domains with a large number of problem instance classes, and it places an undue burden on prompt engineers to provide instance-specific examples. Finally, our experiments call into question claims of enhanced "emergent reasoning" of LLMs with prompt engineering efforts; corroborating contemporary research (Verma et al., 2024; Valmeekam et al., 2024; Stechly et al., 2024b; Ullman, 2023; Schaeffer et al., 2023; McCoy et al., 2023; Stechly et al., 2024a; Sprague et al., 2024) questioning the reasoning abilities of LLMs.

```
                              ReAct
Interact with a household to solve a task. Here are two examples.
<EXAMPLE 1 of same task as QUERY>
You are in the middle of a room ... <Task Description> Your task is to : put some spraybottle on toilet.

Act 1: think : To solve the task, I need to find and take a spraybottle, then put it on toilet.
Obs 1: OK.
Act 2: think : First, I need to find a spraybottle. A spraybottle is more likely to appear in cabinet (1-4),
        countertop (1), toilet (1), sinkbasin (1-2), garbagecan (1).
        I can check one by one, starting with cabinet 1.
Obs 2: OK.
Act 3: go to cabinet 1.
Obs 3: On the cabinet 1, you can see cloth 1, a soapboar 1, a soapbottle 1.
...
Act 6: think : Now I find a spraybottle (2), next I need to to take it.
Obs 6: OK.
...
<EXAMPLE 2 of same task as QUERY>
...
<QUERY>
Here is the task. You are in the middle of the room ... Your task is to put some soapbottle in toilet.
```

- ■ Interleaved Think tag
- ■ Reasoning Trace / Guidance
- ■ Example Task / Query Task

Figure 1: An example of ReAct in AlfWorld. We highlight the main components of ReAct, i.e., Interleaved reasoning and acting, the reasoning trace / plan guidance and the example and query task.

## 2 Related Work

### 2.1 LLMs for Planning & Reasoning

Large Language Models have been shown to be successful in a plethora of natural language tasks (Kocoń et al., 2023; Gilardi et al., 2023; Zhu et al., 2023; Bubeck et al., 2023; Bhattacharjee et al., 2023). However, there are two schools of thought when it comes to utilizing off-the-shelf LLMs for planning and reasoning tasks. Works such as Chain of Thought, ReAct, and others that followed (Wei et al., 2022; Yao et al., 2023; Long, 2023; Yao et al., 2024; Besta et al., 2024; Fu et al., 2024; Aksitov et al., 2023), have argued about the reasoning abilities of LLMs by proposing prompting methods. On the other hand, (Valmeekam et al., 2024; Stechly et al., 2024b) have refuted these claims by showing the inability of LLMs to solve deterministic planning and classical reasoning problems.

### 2.2 ReAct & Anthropomorphizing Reasoning

Lately, works such as ReAct, Reflexion, and their other variants (Yao et al., 2022b; Shinn et al., 2023) have argued on the prowess of LLMs' reasoning abilities on text-based decision-making domains such as AlfWorld Shridhar et al. (2020) and WebShop Yao et al. (2022a). However, the use of generating reasoning traces and interleaving them with task-specific action execution, which is the central claim of ReAct, has been used in many other applications besides these domains. Furthermore, there have been several extensions to ReAct that claim to boost their generalization abilities across more domains including multi-modal domains (Yang et al., 2023; Castrejon et al., 2024), autonomous vehicles (Cui et al., 2024), table question answering (Zhang et al., 2023), etc. While the effectiveness of ReAct is celebrated across different areas, these works only depend on anthropomorphization of LLMs (Min et al., 2022; Peng et al., 2024) for using ReAct-based prompting with no justification on the source of improvement in performance. This motivates our work in investigating the components of ReAct with respect to sequential decision-making problems and analyzing the role each component plays.

## 3 Preliminaries

### 3.1 Domains

**AlfWorld:** (Shridhar et al., 2020) is a synthetic text-based game built on top of a STRIPS-style PDDL domain description (Fikes & Nilsson, 1971). ReAct (Yao et al., 2022b) defines six tasks (or problem classes) within this domain namely - Put, Clean, Heat, Cool, Examine, and PutTwo. Each problem class consists of several problem instances, such as *put a spraybottle on toilet* (see Fig. 1 is an example instance of Put

class. Since AlfWorld is a partially observable environment, each of these problem instances can be solved by navigating and interacting with the environment simulator via text actions. For example, this task can be solved by the following actions- *go to cabinet 2, take spraybottle 2 from cabinet 2, go to toilet 1, put spraybottle 2 in/on toilet 1.*

**WebShop:** (Yao et al., 2022a) is an online shopping website environment with 1.18M real-world products and 12K human instructions. The agent is provided with an initial human instruction (for example, "I am looking for a nightstand with drawers. It should have a nickel finish, and priced lower than $140"). The agent's task is to crawl the shopping environment using actions such as `search 'nightstand drawers'`, `choose 'white buttons'`, `back to search`, etc. For this work, we randomly sample 500 test instructions from the environment and evaluate the success rate of the agent's task completion.

## 3.2 ReAct

ReAct (Yao et al., 2022b) claims to increase LLM's performance on text-based planning tasks such as AlfWorld and WebShop primarily by augmenting the original action space of the agent with a *think* action. The *think* action tag provided by ReAct is claimed to comprise of **Re**asoning + **Act**ion trace that is provided in the solution for the example problems (exemplars) as part of the prompt. During execution, the LLM is expected to generate a *think* action tag for the queried problem instance that is semantically similar to the one provided for the examples in the prompt.

**Location of *THINK* tag:** In ReAct, the integration of the *think* tag within actions serves to expand the action space. This allows the language model (LLM) agent to execute a *think* action, prompting an *'OK'* response. Through analysis of example prompts in ReAct experiments, we identify various instances of the *think* action. Typically, it appears after stating the problem instance, reiterating the task, and providing problem-specific guidance. However, the authors offer no structured guidelines for its implementation, placement, or guidance. This observation aligns with feedback from the paper's reviewers (OpenReview, 2024) citing inconsistencies in the prompting format.

**Content of *THINK* tag:** In ReAct, the *think* action consistently provides the decision-making agent with success-oriented guidance for task completion. For instance, upon encountering a `spraybottle`, the prompt might include: `think: Now I find a spraybottle (2). Next, I need to take it.` This guidance exposes forthcoming actions and sub-tasks for the agent.

**Few shot *EXAMPLE*s:** In the AlfWorld domain (which is a PDDL domain), ReAct authors (Yao et al., 2022b) classify six problem classes or tasks: `Put, Clean, Heat, Cool, Examine, PutTwo`. Despite representing different tasks, they share the same environment dynamics and action space, allowing for very similar execution trace. For instance, a `Heat` task might involve `Putting` an item into a microwave. In ReAct experiments, authors provide two example problem-solution pairs (referred to as exemplars in our work) before querying the LLM agent with a problem instance. Authors force ReAct agent to use examples and queries belonging to the same problem class without motivating this design decision. However, the queried problem may differ in objects or locations from the exemplars.

# 4 Evaluating ReAct Prompting

The subsequent sub-sections explore the design of exemplar prompt variations to investigate our research questions concerning the claims of ReAct. Each variation modifies the base ReAct prompt, and we use the AlfWorld domain as the running example for discussing these variations (see Appendix for WebShop prompts).

## 4.1 RQ1: Interleaving Thinking and Acting - Location of *THINK* tag

*Does the agent performance depend on interleaving reasoning trace with action execution?*

To answer this research question, we propose collating the guidance information contained within the multiple *think* tags present in the examples of the input prompt into a single *think* tag appended after the example

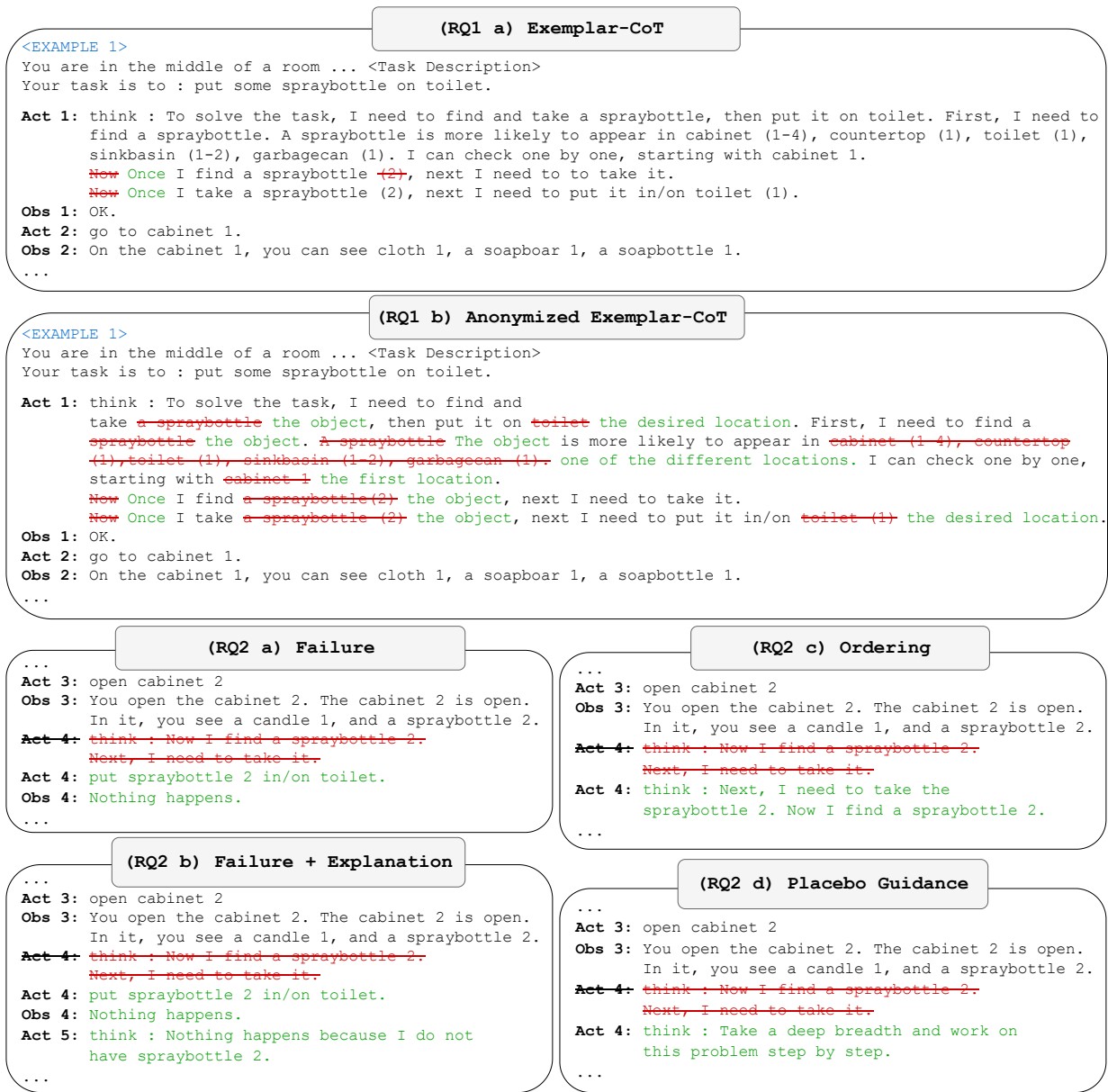

Figure 2: Example of prompt variations considered for RQ1 and RQ2.

problem is specified. This approach can be interpreted as Chain-of-Thought (Kojima et al., 2022; Wei et al., 2022), where guidance information is generated before action execution.

**Variation 1 - Exemplar-based CoT:** AlfWorld is a partially observable environment where an agent can only observe objects after reaching that location. Hence, we remove specific location and object identifiers to modify the *think* actions that are originally interleaved with other actions in the environment (see RQ1 a in Figure 2). Finally, we append all the *think* actions together at the beginning of the example problem. For example, we rewrite it as - `think:` ~~Now~~ `Once I find a spraybottle` ~~(2)~~`, next I need to take it.` **Intuition:** Problem-specific guidance for a sequential decision-making agent can be given step-by-step (as in ReAct) or all at once. Note, that this variation is possible since AlfWorld is not a dynamically changing environment in which case providing information on the task will not be possible.

**Variation 2 - Anonymized Exemplar-CoT:** We take one step further and modify the *think* tag to remove references to specific locations and objects, making it more general (see RQ1 b in Figure 2). Similarly, in WebShop we anonymize specific item options as `desired options`. For example, `think:  To solve the task, I need to find and take` ~~`a spraybottle`~~ `the object,` then put it on ~~`toilet`~~ `the desired location`. **Intuition:** Exemplars can be made more general by providing abstract guidance and exploiting LLMs ability to identify necessary semantic entity relations. This can be useful for a prompt design that can suitably encompass a wider range of downstream tasks and mediate the need of providing problem-specific guidance in the *think* tags.

## 4.2  RQ2: Action Guidance by Thinking - Content of *THINK* tag

*How does the nature of the reasoning trace affect the performance of LLM?*

ReAct claims to use reasoning trace as the guidance information following the *think* tag. For instance, in ReAct (Yao et al., 2022b), thoughts are to (1) decompose the goal (2) track sub-goal completion (3) determine the next sub-goal and (4) reason via common-sense where to find and object and what to do with it. It is, however, unclear what is the motivation to use these as the reasoning trace. The potential anthropomorphization of large language models (LLMs) may suggest that their thought processes are similar to the abstract plans humans make, and that they must be prompted in the same manner. However, it is unclear why this assumption should hold true.

**Variation 1 - Failure:** We note that none of the examples used in ReAct prompting for any task consist of invalid actions. We inject two invalid actions in the execution trace: the first that attempts to execute the action pertinent to the task, such as *put spraybottle 2 in/on toilet*, when not possible and, second, executes some other invalid action (see RQ2 a in Figure 2). We include the expected simulator response, *Nothing happens.*, when invalid actions are taken. **Intuition:** Reasoning trace can be about *what to do* such as future sub-goals, or *what not to do* such as mistakes in hindsight. This should be weaker guidance than in base ReAct as the exemplars do not point out what to do next.

**Variation 2 - Failure + Explanation:** We place *think* actions after invalid actions injected in Failure Variation which consist of explanations for the failure such as, `think:  Nothing happens because I do not have a spraybottle 2` (see RQ2 b in Figure 2). **Intuition:** We can augment mistakes with explanations to avoid similar failures. This is a stronger guidance signal than Failure, however, the exemplars still not provide information on what to do next.

**Variation 3 - Guidance Ordering:** LLMs are known to be susceptible to minor syntactic perturbations to inputs. We test whether it is true for guidance information given as prompt as well (see RQ2 c in Figure 2). We identify chain of sub-tasks in a reasoning trace $S_1 \rightarrow S_2 \cdots S_n$ and reverse it to be $S_n \rightarrow S_{n-1} \cdots S_1$. For instance, `think:  Now I find a spraybottle (2).  Next, I need to take it.` becomes `think:  Next, I need to take the spraybottle (2).  Now I find a spraybottle (2).` **Intuition:** LLM agent should be invariant to the reasoning trace syntax if the semantic information is preserved.

**Variation 4 - Placebo Guidance:** It is unclear to what extent LLM agent uses the supposed helpful thoughts for the decision making task. In this variation we replace *think* tag guidance with a placebo thought that does not contain any task relevant information, such as `think:  Take a deep breath and work on this problem step by step.` (see RQ2 d in Figure 2), but has been widely used as prompt engineering trick (Kojima et al., 2022). **Intuition:** According to claims of ReAct, we expect the performance to get worse when the guidance does not have any information useful for task success.

## 4.3  RQ3: *EXAMPLE*s - *QUERY* Similarity

*How does the similarity between the prompt examples and the query problem affect LLM performance?*

RQ3 investigates the role of example similarity to the query in LLM agent's performance. Establishing problem similarity can be challenging, especially where minor variations to the problem can have varied interpretations (such as an analogy to a different task altogether). Our work explores this challenge in a

systematic way. During example prompt construction, prompt designers may use synonyms to refer to objects (`Domain`), come up with examples where the agent task is the same as query but the goals are different (`Instance`), or provide optimal solutions as the examples (`Optimal`) preventing LLM to obtain information regarding exploration strategy. Furthermore, given that the domain has the same underlying action dynamics and that the tasks reuse several actions, prompt designers may choose to provide query specific example prompts (as in base `ReAct`), provide one of a different task and one of the same task (`One`), provide both examples to be of a different task (`Both`), or take an exhaustive approach and provide one example of all tasks (`All`).

**Variation 1 - Synonyms (`Domain`):** For this variation, we replace the object and location names in the example prompts with their synonyms. For example, *spraybottle* → *aerosolbottle*, *cabinet* → *cupboard*, and, *microwave* → *oven*. We make 36 such changes to object and location names across all the examples.[1] **Intuition:** Exemplar guidance maybe specified with alternate synonymous object and location names. Reasoning agents should be invariant to variable name substitution for closed world dynamics such as PDDL-based AlfWorld. When example-based guidance needs to be provided to a sequential decision making agent, examples may consist of synonymous object and location names. Hence, we expect to achieve similar task performance.

**Variation 2 - Problem Level (`Instance`):** We inject instance-level changes to the examples provided in the prompts. We change the goal location in exemplar problem to ensure that it does not match with any of the goal locations in query problem. We also add repetitive yet futile actions in the exemplar execution trace which does not effect the solution. **Intuition:** Ensuring a different goal location in exemplar from the queried problem is a natural use-case. Moreover, exemplars may contain arbitrary exploration strategies such as action repetition (Sharma et al., 2017).

**Variation 3 - Problem Level (`Both, One, All`):** Recall that the environment dynamics for all the tasks are the same. In fact, several tasks subsume the use of our tasks such as `Heat` requires the agent to `Put` an food in the microwave. In general, all the tasks share a large portion of actions (such as exploring cabinets and locations, picking objects etc.). Motivated by how tight relationship of these tasks we come up with three variations. First, `One`, uses one exemplar of an arbitrarily picked task and the other exemplar of the same task as the query. Second, `Both`, uses both exemplars from an arbitrarily picked task. Finally, `All`, uses a total of six exemplars corresponding to each task under consideration. Note, that while all other prompt variations use two in-context examples, the `All` variation uses six in-context examples. Remember, this includes the query task which is always present at the end in the input prompt. **Intuition:** With a very similar action execution trace (such as exploration, picking and placing objects) across tasks, and shared dynamics, LLM agent should be minimally affected by the use of exemplars of a different task.

**Variation 4 - Exploration Strategy (`Optimal`):** As noted before, ReAct does not explain the choice of exemplars used. An important ingredient to the exemplars is the exploration strategy used. In this variation we provide exemplars which serendipitously take the optimal actions (as if the environment were fully observable) and therefore the example plan is the shortest possible. **Intuition:** Exploration strategy exposed in exemplars (that too for the same problem task) should not impact ReAct's performance if the LLM agent is reasoning instead of retrieval (or pattern matching).

## 5 Results

While the original ReAct experiments were carried out on PaLM (currently decommissioned), we reproduce their results with newer set of models. We use GPT-3.5-Turbo, GPT-3.5-Instruct, GPT-4, GPT-4o, and Claude-Opus, which are all newer models than those benchmarked in ReAct (Yao et al., 2022b). Note, that despite using newer models, our results shed doubts on the reproducibility and consistency across models of the original paper's results. As noted, we use the setup in (Yao et al., 2022b) for all our experiments. In AlfWorld, GPT3.5-(Turbo, Instruct) results are on 134 instances across six tasks, GPT-4/Claude-Opus on 60 instances (10 for each task) due to cost considerations. In WebShop, GPT3.5(Turbo, Instruct), GPT-4o, and LLAMA-3.1-8B results are on 500 samples, while GPT-4/Claude-Opus are on 50 instances due to cost

---

[1]The object names / location are unchanged for the problem query and subsequent interaction with the simulator.

considerations. For each of the experiments, we include the `Act` baseline, originally used by ReAct, where the same examples are provided to prompt the LLMs but without any *think* tags in their execution traces. Naturally, in this case, the LLM is unable to produce a *think* tag for the queried problem, and thus, unable to utilize any supposed reasoning guidance given as part of the in-context examples. We include the lower and upper bound confidence intervals for all our experiments in the Appendix Section B.3.

Table 1: Average Success % of LLM for RQ1 and RQ2 on six **AlfWorld** tasks. The table uses a color spectrum to represent the quality of the numbers. Cells shaded in green indicate the best performance compared to base ReAct, transitioning to red for the worst relative performance.

| Model / Prompt | Act | ReAct | RQ1 | | RQ2 | | | |
|---|---|---|---|---|---|---|---|---|
| | | | CoT | Anon. CoT | Placebo | Order | Failure | Explanation |
| GPT-3.5-Turbo | 34.3 | 27.6 | 46.6 | 41 | 30 | 28.3 | 43.3 | 41.6 |
| GPT-3.5-Instruct | 44 | 50.7 | 61.9 | 50.7 | 41 | 42.5 | 47 | 44.7 |
| GPT-4-0314 (Old) | - | 23.3 | 43.3 | 33.3 | 36.6 | 30 | 50 | 36.6 |
| GPT-4-0613 (Latest) | 70.0 | 26.7 | 40.0 | 26.6 | 36.6 | 30 | 60 | 36.6 |
| Claude-Opus | 43.3 | 56.6 | 50 | 46.6 | 30 | 50 | 53.3 | 30 |

Table 2: Average Success % of LLM for RQ1 and RQ2 on **WebShop** tasks. The table uses a color spectrum to represent the quality of the numbers. Cells shaded in green indicate the best performance compared to base ReAct, transitioning to red for the worst relative performance.

| Model / Prompt | Act | ReAct | RQ1 | | RQ2 | | |
|---|---|---|---|---|---|---|---|
| | | | CoT | Anon. CoT | Placebo | Failure | Explanation |
| GPT-3.5-Turbo | 1.12 | 1.04 | 2.20 | 1.88 | 1.52 | 3.48 | 3.48 |
| GPT-3.5-Instruct | 7.24 | 7.16 | 7.52 | 6.12 | 7.40 | 7.20 | 7.24 |
| GPT-4-0613 (Latest) | 8 | 4 | 8 | 8 | 6 | 8 | 8 |
| GPT-4o | 4.64 | 2.24 | 4.68 | 4.52 | 4.08 | 4.68 | 4.68 |
| Claude-Opus | 4 | 4 | 4 | 2 | 4 | 2 | 4 |
| LLAMA-3.1-8B | 1.44 | 3.16 | 3.28 | 3.92 | 2.04 | 1.20 | 2.16 |

## 5.1 Utility of Interleaving *THINK* tags (RQ1)

To recall, we create two prompt variations for RQ1: `CoT` and `Anon. CoT`. In `CoT`, we concatenate all the *think* tag content at the beginning of the execution trace for the two examples. For `Anon. CoT`, we further anonymize the object and location names in the concatenated *think* tag.

From Table 1 (RQ1), we note that the `CoT` and the `Anon. CoT` variations perform significantly better than base ReAct for all OpenAI models. Moreover, the performance dips slightly for Claude-Opus along these variations. This refutes ReAct's first claim on the importance of interleaving reasoning trace generation with action execution. Even in the case of Claude where there is a slight dip in performance, the models seems to be performing at reasonably high success rate which questions the importance of interleaved reasoning and action execution. We omit LLAMA-3.1-8B and GPT-4o (See B.2) for AlfWorld as they achieve zero performance over baselines and all the variations. From Table 2, we find a similar pattern: `CoT` and `Anon. CoT` variants perform closely or better than the baseline ReAct. A surprising result consistent in both the domains is the performance of `Act` method. `Act` baseline is weaker only for two models `GPT-3.5-Instruct, Claude-Opus` for both the domains, which further questions the utility of using ReAct prompting.

### 5.1.1 RQ1 Failure Analysis

Figure 3 shows a radar chart for four different LLMs, namely GPT-3.5-Turbo, GPT-3.5-Instruct, GPT-4, and Claude Opus, for the RQ1 ablations indicating the failure rate (in %) across the six tasks in the AlfWorld domain. In all but one case for Claude Opus, we note the largest shaded region in red indicating the highest failure rate for ReAct across five AlfWorld tasks as compared to the RQ1 prompt variations. This indicates

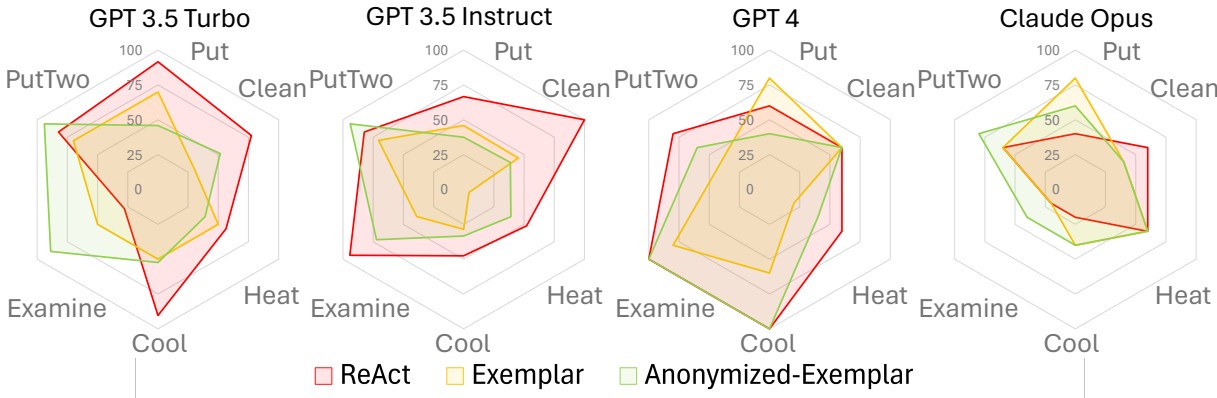

Figure 3: **RQ1 Failure Analysis:** The radar chart shows the failure rates of various LLMs with different ReAct-based prompt settings for RQ1 (Base React, Global, Anonymized) across six AlfWorld tasks (hexagon vertices). Higher values / larger shaded region indicate worse performance.

that there is no performance gain from interleaving *think* tags with actions in the prompt examples, rather has a contrary effect on majority of the tasks. `CoT` has the smallest average failure rate, followed by the `Anonymized CoT`. We believe that this is because prompting LLMs with object and location-specific guidance as part of *think* tags in the in-context demonstrations provides more useful and targeted information for solving the downstream queried problem, which too has similar object and location identifiers as the in-context examples. We have also included a direct comparison of the *think* tags generated by the ReAct framework and by the RQ1 prompt variations in Appendix Section B.4.

## 5.2 Utility of *THINK* tag Content (RQ2)

To recall, we create four prompt variations for RQ2: `Placebo`, `Failure`, `Explanation` and `Order`. In `Placebo`, we replace each *think* tag content with meaningless text that has no relation to the task. For `Failure` and `Explanation` variations, we replace the *think* tag suggested actions to an action that does not help solve the task, along with an explanation for the same, respectively. In `Order`, we switch the ordering of the sentences that are part of each individual *think* tag.

Recall that reasoning trace guidance pertains to the prospective actions or behaviors an agent should execute (foresight guidance). This type of guidance is more informative compared to other variations, such as hindsight guidance, which focuses on past errors without providing future solution steps, and placebo guidance, which is entirely unrelated to the task. ReAct claims that reasoning trace is crucial for LLM agent performance, which would predict a decline in performance with hindsight guidance and a collapse with `Placebo` variation. Therefore, a practitioner would expect all the rows to be a dark shade of `red`. In contrast, our findings in Table 1 indicate that hindsight guidance (`Failure`, `Explanation`) actually improves the performance of the GPT family of models. The Claude-Opus model's performance remains stable with hindsight (`Failure`) guidance and declines with `Placebo` guidance. This refutes ReAct's claim that task-specific reasoning trace is the source of LLM performance. Our argument that LLM agent's performance is only slightly affected by the reasoning trace explains the indifference to `Order` perturbation as well. If the LLM is not utilizing the reasoning trace for decision making, change in ordering would not affect the agent's performance. Our arguments hold for the WebShop domain as well, where all of the variants perform closely or better than the baseline ReAct.

Finally, contrary to the general perception that better GPT models would improve over reasoning, we find that GPT-4-(Old)'s performance is the worst among GPT-X family further highlighting the brittleness of claims of ReAct. GPT-4-(Latest) performs similarly to GPT-4-(Old), except for the Act baseline which again shows the futility of ReAct prompting. In all our experiment settings, we note that LLMs replicate the exact

Table 3: Average Success % of LLM for RQ3 on six AlfWorld tasks. OC: Out of context limit. The table uses a color spectrum to represent the quality of the numbers. Cells shaded in green indicate the best performance compared to base ReAct, transitioning to red for the worst relative performance.

| Model / Prompt | Act | ReAct | RQ3 | | | | | |
|---|---|---|---|---|---|---|---|---|
| | | | Domain | Instance | Optimal | All | One | Both |
| GPT-3.5-Turbo | 34.3 | 27.6 | 1.6 | 30 | 20.1 | 32 | 28.3 | 1.6 |
| GPT-3.5-Instruct | 44 | 50.7 | 47.6 | 42.5 | 39.5 | OC | 17.9 | 5.2 |
| GPT-4-0314 (Old) | – | 23.3 | 13.3 | 23.3 | 50 | 23.3 | 16.6 | 0 |
| GPT-4-0613 (Latest) | 70.0 | 26.7 | 10.0 | 20.0 | 53.3 | 23.3 | 20 | 3.3 |
| Claude-Opus | 43.3 | 56.6 | 50 | 46.6 | 43.3 | 50 | 60 | 6.6 |

steps as shown for the examples in the prompts. Hence, they do not output what ReAct claims as think tags if those tags are not present in the original prompt.

### 5.2.1 RQ2 Failure Analysis

Figure 4 shows a radar chart for four different LLMs, namely GPT-3.5-Turbo, GPT-3.5-Instruct, GPT-4, and Claude Opus, for the RQ2 ablations indicating the failure rate (in %) across the six tasks in the AlfWorld domain. Similar to Figure 3, we note the highest failure rate for ReAct across five AlfWorld tasks as compared to the other RQ2 prompt variations, with the smallest average failure rate for Failure prompt variation. We hypothesize that this is because providing actions that lead to failure in the in-context demonstrations assist in preventing similar failure-leading actions for the queried problem. Moreover, providing failure information can be less cognitively demanding for prompt designers than providing an exact solution for the in-context examples, as required by the ReAct framework. We have also included a direct comparison of the *think* tags generated by the ReAct framework and by the RQ2 prompt variations in Appendix Section B.4.

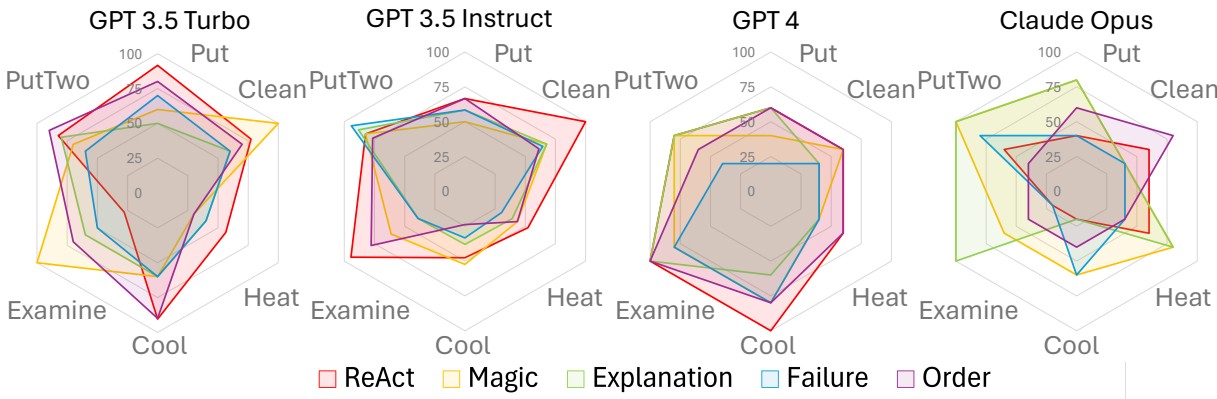

Figure 4: **RQ2 Failure Analysis:** The radar chart shows the **failure rates** of various LLMs with different ReAct-based prompt settings for RQ2 (Base React, Magic, Failure, Failure+Explanation, Ordering) across six AlfWorld tasks (hexagon vertices). Higher values / larger shaded region indicate worse performance.

### 5.3 Dependence on Literal Example-Query Similarity (RQ3)

To recall, we create six prompt variations for RQ3: `Domain`, `Instance`, `Optimal`, `All`, `One` and `Both`. In `Domain`, we use synonymous object and location names. For *Instance*, we add futile but repetitive actions while in `Optimal` we only include the minimum number of actions needed to reach to the goal in the examples. For `All`, we include one example from each of the six tasks. We change one example task to be different from the same category as of the query task in `One` and change both the example tasks from another category in `Both`.

Intuitively, the similarity of `Domain` examples is closest with base ReAct, followed by `Instance` and `Optimal` variations. Finally, `All` contains an overload of information followed by `One` and `Both` which have the same action space but uses different tasks as exemplars. Recall that AlfWorld being a PDDL domain has a shared environment dynamics across all tasks with up to 80% of actions shared across execution traces. While ReAct does not investigate impact of varied exemplars, given the popular usage, one expects LLMs to be robust to such changes especially in a common-sense household domain. Table 3 shows the severe brittleness of ReAct based LLM agent to even minor variations (such as `Domain, Instance`). Specifically, performance of GPT-3.5-Turbo and GPT-4 plummets for `Domain`. Claude-Opus which was more robust in RQ1, RQ2, is also impacted severely by `Domain, Instance` variations. Furthermore, the performance of all LLMs but GPT4 drops when we do not expose the exploration strategy and only provide `Optimal` exemplars.

Overloading the LLMs with more exemplars `All` does not impact its performance. We posit, this is because the query-task exemplar is still part of the large input prompt. Among the two exemplars, as provided in ReAct, when one of them is of a different task (`One`) then the performance significantly reduces for LLMs. When both of the exemplars are of a different task then the performance collapses to single digit success rates for all the models. This is a key result of this work highlighting the severe dependence of LLMs on the similarity of the exemplars to the query task and the lack of generalization with ReAct. Through sensitivity analysis using our RQ3 variations, we could find parts of the input (the task similarity of the exemplar with query) which is the source of ReAct performance. Essentially, the LLM is mimicking/performing approximate retrieval from the context presented to it by trying to overfit on the information provided as part of the in-context demonstrations (or examples). Moreover, our results corroborate the line of research that questions the inability of LLMs to reason or plan (Verma et al., 2024; Valmeekam et al., 2024; Stechly et al., 2024b; Ullman, 2023; Schaeffer et al., 2023; McCoy et al., 2023; Stechly et al., 2024a; Sprague et al., 2024).

The reported success-rate from the ReAct paper Yao et al. (2022b) on the WebShop domain is 40%. Due to the absence of the exact queries used in the paper, we randomly sampled queries from the WebShop dataset comprising 12K records. This approach possibly resulted in the decoupling of any relationship between the exemplars and the queries. Referring to Table 2, it is evident that the performance of the WebShop ReAct agent significantly declined, reaching single digit percentages (as well as other variants). This mirrors the trends observed in the `Both` variant of the Alfworld in Table 3, further supporting our arguments.

### 5.3.1 RQ3 Failure Analysis

Figure 5 shows a radar chart for four different LLMs, namely GPT-3.5-Turbo, GPT-3.5-Instruct, GPT-4, and Claude Opus, for the RQ3 ablations indicating the failure rate (in %) across the six tasks in the AlfWorld domain. We note the highest failure rate for the Both variation across five AlfWorld tasks as compared to the other RQ3 prompt variations, with the smallest average failure rate for Optimal prompt variation. These results further indicate the brittle dependence of ReAct prompting on the type of in-context demonstrations used for querying the LLM for downstream tasks. We have also included a direct comparison of the *think* tags generated by the ReAct framework and by the RQ3 prompt variations in Appendix Section B.4. We also did a preliminary study on the similarity (computed in terms of embedding distances) between the 134 prompting tasks with the originally provided examples, and the performance of ReAct. As can be seen from Figure 6 in Appendix Section B.6, as expected, the performance degrades significantly in the off-diagonal cases.

### 5.4 Length & Inter-task Generalization

**Unrolling and Inter-task Similarity:** We perform additional experiments to highlight the lack of length generalization where the query task is to essentially repeat the task in the exemplar (*Unrolling*). For instance, in AlfWorld, the exemplar is `Put` and the query is `PutTwo` to put two objects at given location. In this case, the LLM has to unroll the given advice and repeat exemplar task execution to solve the query. The success rate of GPT-3.5-Instruct (the best performing GPT model in our experiments) drops down from 52% to 9%. Similarly, we experiment with a *Inter-task Similarity* variation where the exemplar task subsumes execution of the query task. For instance, the `Heat` task requires the agent to pick and place object in the microwave (which is an instantiation of `Put` task). One would expect that `Heat` is a good exemplar for `Put`, however, the

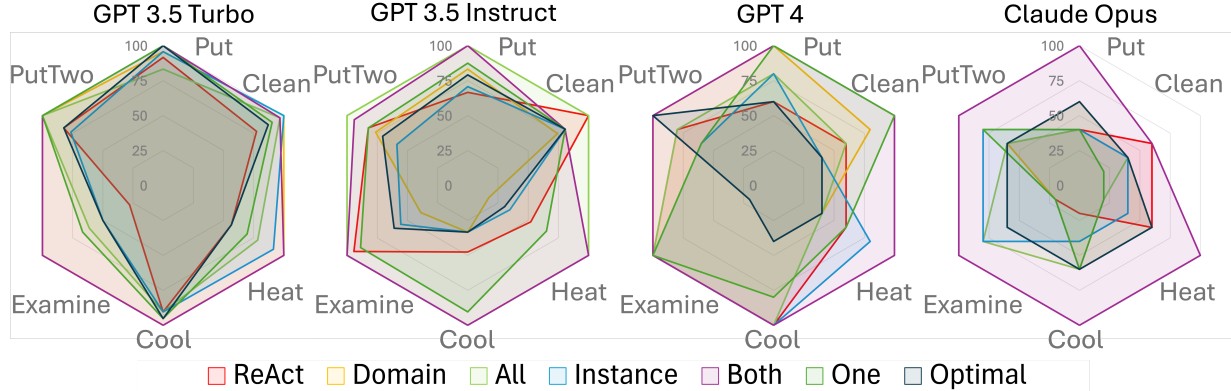

Figure 5: **RQ3 Failure Analysis:** The radar chart shows the **failure rates** of various LLMs with different ReAct-based prompt settings for RQ3 (Base React, Domain, Instance, All, Both, One) across six AlfWorld tasks (hexagon vertices). Higher values / Larger shaded region indicate worse performance.

performance of GPT-3.5-instruct model goes from 18% to 0% in this case. These results further underscore the brittleness and the need for instance-specific exemplars in ReAct.

**Exemplar CoT prompt from RQ1 + RQ3 variations:** In this case, we test the main results of our work with the RQ1 `CoT` variation to identify whether our findings hold true there as well or not. That is, we test `RQ3-One` and `RQ3-Both` with Exemplar CoT prompt variation. For GPT-3.5-Turbo, we find that the average performance drops from 46.6% (RQ3- `CoT`) as in Table 1 to 28.3% in `One` and 10.4% in `Both` variation cases, and remains at 40.3% for `All` variation, thereby reaffirming our hypothesis behind the lack of generalization in ReAct-based LLM prompting framework.

# 6 Discussion

**Operationalizing '*think*' actions by LLMs** Given the free-form nature of *think* tag generation and arbitrary nature of their content (about sub-task, common-sense next steps etc.), checking whether the generated thoughts are in-fact reasonable is a challenging problem. For completeness, we find that 40% of the times after generation of a *think* tag, subsequent environment action taken by the LLM was invalid (for GPT-3.5-instruct) in AlfWorld. It is much higher ( 80% for GPT-3.5-Turbo,  90% for Claude-Haiku) for weaker LLM models. This further highlights the inability of LLMs to operationalize its generated *think* action, as also seen in (Roy et al., 2024). From manual inspection we find that the typical *think* action would enlist all possible locations as next locations to visit for most of the tasks. As demonstrated in Section 5.2, the performance of LLMs actually decreases when provided with foresight guidance, as seen with the base ReAct model.

**Lack of Generalization in ReAct-Style Prompting:** Recall, that ReAct claims an improved performance for text-based planning domains, namely - AlfWorld and WebShop, where the presence of a *think* tag acts as guidance for the LLM to generate the next set of actions during the LLM-environment interaction. Through our sensitivity analysis, we dissect each component of ReAct-style prompting in a critical effort to understand the factor that leads to the observed success rates in these domains. With variations on the placement (RQ1) and content (RQ2) of the *think* tag, we eliminate it as the primary cause of any improvement. Furthermore, slight variations in exemplar tasks (RQ3) lead to a stark decline in success rate, clearly indicating the literal dependence of performance on the highly curated instance-specific examples provided by domain experts. While some works on In-Context Learning point out the impact of well-curated examples (Min et al., 2022; Peng et al., 2024), our work specifically highlights the severe lack of generalization and that exemplar-query similarity is the primary cause of ReAct's performance, thereby rejecting contemporary belief that the heavy-lifting of LLM reasoning & planning is done through the *think* tag.

**Lack of Generalization to newer LLMs:** ReAct uses the `Act` baseline (LLM-environment interaction without *think* tags) in their work to showcase improvements due to the presence of the proposed *think* tag. For AlfWorld, ReAct reports 45% success rate for `Act` baseline and 71% success rate for `ReAct` prompting using the PaLM model. For WebShop, ReAct reports 30.1% success rate for `Act` baseline and 40% success rate for `ReAct` prompting on PaLM. However, we note from our results on both domains that the `Act` baseline performs much better than ReAct for several LLMs, which questions the compatibility of ReAct to newer-age LLMs. ReAct performs worse with newer models as compared to the results they report on the currently decommissioned PaLM. Moreover, this further strengthens our claim that ReAct's performance can't be contributed to the presence of *think* tags, but is primarily due to the literal similarity between the in-context examples and the queried problem. This observation also questions the contemporary belief that such prompting strategies are generalizable throughout different LLM families, including newer models.

We re-iterate our key result, given any LLM model, the success rates plummet with our RQ3 variations showing a consistent pattern of literal dependence on the provided examples irrespective of the LLM. Moreover, the performance of all the LLMs remain quite high (if not better) when we vary the location and content of the *think* tags. This highlights the need for higher rigor in agentic LLM experimentation and in-depth evaluation seeking source of improvements.

## 7 Conclusion & Future Directions

ReAct based prompt engineering methods have been claimed to improve planning abilities of Large Language Models. In this study, we critically examine ReAct along three dimensions, informed by its claims and our hypotheses regarding its performance sources. Contrary to ReAct's claims, our findings reveal that its performance is neither due to interleaving reasoning trace and guidance information generation with action execution, nor due to the specific nature of the guidance information. Instead, we identify that the true source of LLM performance in sequential decision-making tasks, such as AlfWorld, is the high degree of literal similarity between exemplar problems (few-shot) and the query task, thereby leading to severe lack of generalization. Our findings caution against an uncritical adoption of ReAct-style frameworks for their putative abilities to enhance performance in domains requiring planning. To conclude, we believe that these results will be helpful for practitioners and future works, particularly when designing prompts for decision-making problems, and benefit from avoiding putting any efforts into constructing non-trivial problem-specific reasoning traces.

## Broader Impact

With the surge in LLM-based applications, including those that may have critical consequences, it becomes imperative to study the robustness of LLMs and frameworks based on them. One such case in point is that of ReAct, which has gained popularity for its use across diverse domains. However, there has been little to no critical analysis to understand the robustness and generalizability in the ReAct framework. These include, but are not limited to, understanding the core reasons behind ReAct's performance, performance of ReAct on newer LLMs, robustness to slight deviations from the intended problems, etc. Our work presents a critical sensitivity analysis of the framework to answer these questions via a comprehensive set of empirical results, thereby revealing the brittleness of the proposed framework along with the over-reliance on human-provided task-specific examples. Ideally, the impetus of providing generalization and robustness guarantees, if any, should fall on the authors of the proposed approach. This would allow future works to have a thorough understanding of various approaches and utilize them responsibly.

## Acknowledgment

This research is supported in part by ONR grant N0001423-1-2409, DARPA grant HR00112520016, and gifts from Qualcomm, J.P. Morgan and Amazon.

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

# A    Resources Used

In this work we leverage OpenAI API and Claude API for prompting the Language Models.    We use `gpt-4-0613` for GPT4, `gpt-3.5-turbo-0125`, `gpt-3.5-turbo-instruct`, `claude-3-opus-20240229`, `claude-3-sonnet-20240229` and `claude-3-haiku-20240307` for AlfWorld in April-May 2024. Results using `GPT-4-0613` (for both AlfWorld and WebShop) and all the results on WebShop were completed in Sep 2024. As an estimate, for AlfWorld, ReAct and corresponding experiments use approximately 14M input tokens (due to repeated prompting after each action execution) and 150K output tokens for 134 problem instances as used by ReAct.

# B    Additional Considerations

## B.1    Performance of Claude-Haiku

We skip on mentioning the performance of Claude-Haiku, since it was not able to generate syntactically correct actions for any of the instances. We found that following our instruction to generate specific actions as in the exemplar was difficult. We improved the prompt to have specific instructions for generating actions (See D.3 ) but it did not yield any improvements for Claude-Haiku. However, the instruction did help with Claude-Sonnet and Claude-Opus. We find that Claude-Sonnet follows a similar pattern as GPT-3.5-Instruct as presented in our results, and decided to focus ourselves on the strongest/largest Claude model (Claude-Opus) for our evaluation.

## B.2    Performance of GPT-4o and LLAMA-3.1 in AlfWorld

In our experiments, we found that GPT-4o obtained zero success-rate across all variations (including Act and ReAct and our proposed variations). Upon closer inspection, we found that GPT-4o requires significant effort in instruction tuning for AlfWorld, specifically, it would start emitting justifications for why a previously taken action was unsuccessful rather than generate syntactically accepted `think` tags and environment actions. Even for action generation, GPT-4o would pre-pend the actions with justifications, thereby expecting the users to write specific parsers. While stronger parsers maybe possible to implement, we hold GPT-4o to the same expectations as other models (which do not get the benefit of stronger parsers) and report our findings on a consistent evaluation.

LLAMA-3.1-8B obtains zero success-rate across all variations in AlfWorld as well.  We observe that the LLAMA model would generate incorrect actions and would repeat those actions exhausting the iteration budget.

## B.3    Confidence Intervals

In this subsection, we present the experiment results on Alfworld and WebShop with the confidence intervals computed with 95% confidence level. The results have been discussed in Tables 4, 5 and 6.

Table 4: Lower & Upper bound Success % of LLM for RQ1 and RQ2 on six **AlfWorld** tasks with 95% confidence interval. The table uses a color spectrum to represent the quality of the numbers. Cells shaded in green indicate the best performance compared to base ReAct, transitioning to red for the worst relative performance.

| Model / Prompt | Act | ReAct | RQ1 | | RQ2 | | | |
|---|---|---|---|---|---|---|---|---|
| | | | CoT | Anon. CoT | Placebo | Order | Failure | Explanation |
| GPT-3.5-Turbo | 34.3 | 27.6 | (36.8, 56.4) | (31.4, 50.6) | (21.0, 39.0) | (19.5, 37.1) | (33.6, 53.0) | (31.9, 51.3) |
| GPT-3.5-Instruct | 44 | 50.7 | (52.4, 71.4) | (40.9, 60.5) | (31.4, 50.6) | (32.8, 52.2) | (37.2, 56.8) | (35.0, 54.4) |
| GPT-4-0314 (Old) | - | 23.3 | (33.6, 53.0) | (24.1, 42.5) | (27.2, 46.0) | (21.0, 39.0) | (40.2, 59.8) | (27.2, 46.0) |
| GPT-4-0613 (Latest) | 70.0 | 26.7 | (30.4, 49.6) | (17.9, 35.3) | (27.2, 46.0) | (21.0, 39.0) | (50.4, 69.6) | (27.2, 46.0) |
| Claude-Opus | 43.3 | 56.6 | (40.2, 59.8) | (36.8, 56.4) | (21.0, 39.0) | (40.2, 59.8) | (43.5, 63.1) | (21.0, 39.0) |

Table 5: Lower & Upper bound Success % of LLM for RQ1 and RQ2 on **WebShop** tasks with 95% confidence interval. The table uses a color spectrum to represent the quality of the numbers. Cells shaded in green indicate the best performance compared to base ReAct, transitioning to red for the worst relative performance.

| Model / Prompt | Act | ReAct | RQ1 | | RQ2 | | |
| --- | --- | --- | --- | --- | --- | --- | --- |
| | | | CoT | Anon. CoT | Placebo | Failure | Explanation |
| GPT-3.5-Turbo | 1.12 | 1.04 | (0, 5.1) | (0, 4.5) | (0, 3.9) | (0, 7.1) | (0, 7.1) |
| GPT-3.5-Instruct | 7.24 | 7.16 | (2.4, 12.7) | (1.4, 10.8) | (2.3, 12.5) | (2.1, 12.3) | (2.2, 12.3) |
| GPT-4-0613 (Latest) | 8 | 4 | (2.7, 13.3) | (2.7, 13.3) | (1.3, 10.7) | (2.7, 13.3) | (2.7, 13.3) |
| GPT-4o | 4.64 | 2.24 | (0.5, 8.8) | (0.4, 8.6) | (0.2, 8.0) | (0.5, 8.8) | (0.5, 8.8) |
| Claude-Opus | 4 | 4 | (0.2, 7.8) | (0, 4.7) | (0.2, 7.8) | (0, 4.7) | (0.2, 7.8) |
| LLAMA-3.1-8B | 1.44 | 3.16 | (0, 6.8) | (0.1, 7.7) | (0, 4.8) | (0, 3.3) | (0, 5.0) |

Table 6: Lower & Upper bound Success % of LLM for RQ3 on six AlfWorld tasks with 95% confidence interval. OC: Out of context limit. The table uses a color spectrum to represent the quality of the numbers. Cells shaded in green indicate the best performance compared to base ReAct, transitioning to red for the worst relative performance.

| Model / Prompt | Act | ReAct | RQ3 | | | | | |
| --- | --- | --- | --- | --- | --- | --- | --- | --- |
| | | | Domain | Instance | Optimal | All | One | Both |
| GPT-3.5-Turbo | 34.3 | 27.6 | (0, 4.1) | (21.0, 39.0) | (12.2, 28.0) | (22.9, 41.1) | (19.5, 37.1) | (0, 4.1) |
| GPT-3.5-Instruct | 44 | 50.7 | (37.8, 57.4) | (32.8, 52.2) | (29.9, 49.1) | OC | (10.4, 25.4) | (0.8, 9.6) |
| GPT-4-0314 (Old) | – | 23.3 | (6.6, 20.0) | (15.0, 31.6) | (40.2, 59.8) | (15.0, 31.6) | (9.3, 23.9) | (0, 0) |
| GPT-4-0613 (Latest) | 70.0 | 26.7 | (4.1, 15.9) | (12.2, 27.8) | (43.5, 63.1) | (15.0, 31.6) | (12.2, 27.8) | (0, 6.8) |
| Claude-Opus | 43.3 | 56.6 | (40.2, 59.8) | (40.2, 59.8) | (33.6, 53.0) | (40.2, 59.8)50 | (50.4, 69.6) | (1.7, 11.5) |

## B.4 Analysis of generated *think* tags

**Example:**

Your task is to: find some apple and put it in sidetable.

***think* tags for example:**

> *think*: To solve the task, I need to find and take an apple, then put it in sidetable.

> *think*: First I need to find an apple. An apple is more likely to appear in fridges (1), diningtables (1-3), sidetables (1), countertops (1), sinkbasins (1), garbagecan (1). I can check one by one, starting with fridge 1.

> ...

> *think*: Now I find an apple (3). Next, I need to take it.

> ...

> *think*: Now I find an apple (3). Next, I need to put it in/on sidetable 1.

> ...

**Query:**

Your task is to: put a pencil in shelf.

**ReAct *think* tags:**

> *think*: To solve the task, I need to find and take a pencil, then put it in a shelf.

> *think*: First I need to find a pencil. A pencil is more likely to appear in a desk (1-2), drawer (1-6), shelf (1-6). I can check one by one, starting with desk 1.

> ...

> *think*: Now I have the pencil (1). Next, I need to put it in a shelf.

> ...

> *think*: The shelf 1 is empty, so I need to find another shelf to put the pencil in. I will try shelf 2.

> ...

> *think*: The shelf 2 already has a mug on it, so I need to find another empty shelf to put the pencil in. I will try shelf 3.

> ... (multiple *think* tags leading to failure)

> *think*: It appears that the pencil is not being accepted in any of the locations in the room. I may need to reconsider the task or find a different approach to solve it.

> *think*: I will try putting the pencil on the desk 2, since it is the only location left that I haven't tried yet.

> **Result**: *FAILURE*

**RQ1: CoT variation *think* tags:**

> *think*: To solve the task, I need to find and take a pencil, then put it in shelf. First I need to find a pencil. A pencil is more likely to appear in desk (1-2), drawer (1-6), shelf (1-6). I can check one by one, starting with desk 1. Once I find a pencil, next I need to take it. Once I take a pencil, next I need to put it in/on shelf.

> ...

> **Result**: *SUCCESS*

**RQ1: Anonymous CoT variation *think* tags:**

> *think*: To solve the task, I need to find and take the object, then put it on the desired location. First I need to find the object. The object is more likely to appear in one of the different locations. I can check one by one, starting with the first location. Once I find the object, next I need to take it. Once I take the object, next I need to put it in/on the desired location.

> ...

> **Result**: *SUCCESS*

**RQ2: Failure + Explanation variation *think* tags:**

> *think*: Nothing happens because I am not at shelf 1.

> ...

> *think*: Nothing happens because I already have a pencil in shelf 1.

> ...

> *think*: Nothing happens because I am not at desk 1.

> ...

> **Result**: *SUCCESS*

**RQ3: Both variation *think* tags:**

> *think*: To solve the task, I need to find a pencil and put it in a shelf. A pencil is more likely to appear in a drawer (1-6), desk (1-2), shelf (1-6), garbagecan (1), laundryhamper (1). I can check one by one, starting with drawer 1.

> ...

> *think*: I didn't find a pencil in drawer 1. I should check the other drawers. Next, I will check drawer 3.

> ...

> *think*: I found a pencil (1) on desk 1. Next, I need to put it in a shelf. A shelf is more likely to appear in shelf (1-6). I can check one by one, starting with shelf 1.

> ...

> *think*: I didn't find a place to put the pencil in shelf 1 or shelf 2. Next, I will check shelf 3.

> ...

> *think*: It seems like there is no specific item in the room to place the pencil. I may need to reconsider the task and find a different approach. Let me review the task again and see if there are any other clues or hints that I may have missed.

> ...

> *think*: It seems like there are no household members available to assist me at the moment. I may need to come back to this task later or try a different approach.

> ...

> **Result**: *FAILURE*

**RQ3: One variation *think* tags:**

> *think*: To solve the task, I need to find and take a pencil, then put it in a shelf. First, I need to find a pencil. A pencil is more likely to appear in a desk (1-2), drawer (1-6), garbagecan (1), or safe (1). I can check one by one, starting with desk 1.

> ...

> *think*: Now I have the pencil (1). Next, I need to find a shelf to put it in. A shelf is more likely to appear in shelf (1-6). I can check one by one, starting with shelf 1.

> ...

> *think*: It seems that shelf 1 is empty. I should try putting the pencil in a different shelf. Let me try shelf 2.

> ...

> *think*: It appears that none of the shelves are the correct place to put the pencil. I should reconsider where to place it. Let me try putting the pencil in the drawer instead.

> ...

> *think*: It seems that none of the locations in the room are suitable for putting the pencil. I may need to reconsider the task and think of a different approach. Let me try putting the pencil back on the desk where I found it originally.

> ...

> *think*: It seems that the task cannot be completed in this room. I may need to reassess the instructions or seek further guidance to successfully complete the task.

> ...

> **Result**: *FAILURE*

## B.5   Additional WebShop Experiment

We run an additional set of experiments with GPT-4o model on WebShop, as shown in Table 7 on 500 sample problems. We once again note that the performance with the newer models as compared to the PaLM model originally used by ReAct is much worse. We also note a similar trend in the relative performance results with our prompt variations against the original ReAct performance, thereby emphasizing the brittleness of the ReAct framework's primary claims.

### B.6   Task Similarity analysis and Performance Comparison

We did a preliminary study on the similarity (computed in terms of embedding distances using the 'MiniLM-L6-v2' model embeddings and computing the cosine similarity between them) between the 134 prompting tasks with the originally provided examples, and the performance of ReAct. As can be seen from Figure 6, as expected, the performance degrades significantly in the off-diagonal cases.

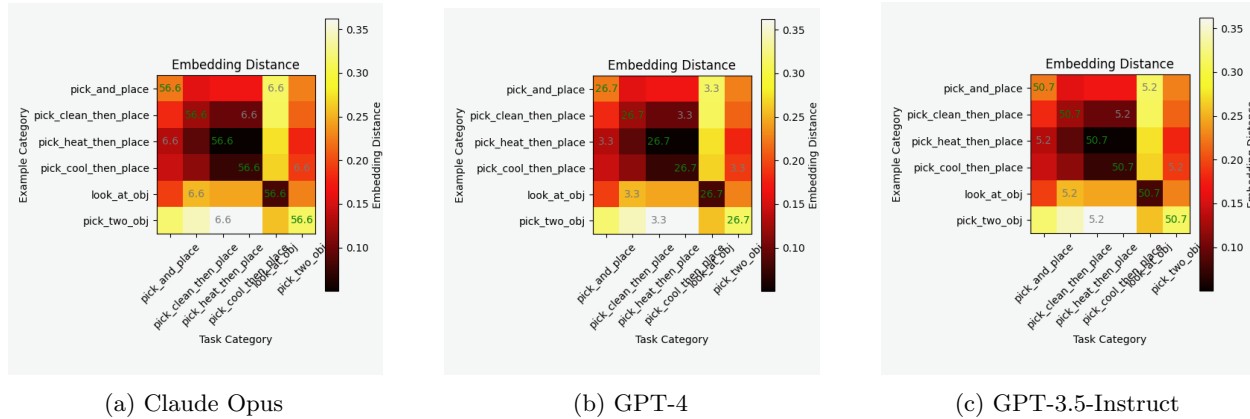

| (a) Claude Opus | (b) GPT-4 | (c) GPT-3.5-Instruct |

Figure 6: Performance Tradeoff with Task Description Similarity in AlfWorld

Table 7: Average Success % of GPT-4o for RQ1 and RQ2 on **WebShop** tasks. The table uses a color spectrum to represent the quality of the numbers. Cells shaded in green indicate the best performance compared to base ReAct, transitioning to red for the worst relative performance.

| Model / Prompt | Act | ReAct | RQ1 | | RQ2 | | |
|:---:|:---:|:---:|:---:|:---:|:---:|:---:|:---:|
| | | | CoT | Anon. CoT | Placebo | Failure | Explanation |
| GPT-4o | 8% | 4% | 9% | 8% | 7% | 8% | 5% |

## C   Experiment Design

Each of the variations proposed along RQ1, RQ2 and RQ3 modifies the few-shot examples only. Remaining aspects such as the query problem or the interaction with the simulator is directly inherited from the ReAct code-base Yao et al. (2022b) at publicly available at `https://github.com/ysymyth/ReAct`. Our code can be found in the attached supplementary material.

Except `All` RQ3 variation, all other settings use the standard two exemplars for prompting the LLM. Depending on the variation we change the content of the exemplar. Full prompts can be found in the attached supplementary code.

### C.1   Running the experiments

In our experiments, according to the variation style we take the exemplar prompts and use the same exemplar prompts across the instances of the query task. Other than `RQ3-Both, One` we use the exemplar of the same task as the query as done in ReAct (and still find brittleness of ReAct). For `RQ3 - Both, One` we use exactly two exemplars but of a different task than query. Finally, `RQ3-All` is the only variation that provides six exemplars (instead of two) and we force the exemplar of the query-task to be appended at the end in the prompt. This was the best performing prompting strategy (on GPT-3.5-Turbo) than when the query-task exemplar was placed at the beginning, at position 4 (middle) and at the end.

### C.2 Hyperparameters

We use $temperature = \tau = 0$ for all of the GPT and Claude models and set `max-tokens = 100` which is borrowed from ReAct's hyperparameters. Rest of the parameters are kept to be default as specified in the respective model's API documentation.

## D   Example Prompts

The full list of curated variations can be found in the supplementary materials. However, for completeness we reference the prompt used for base ReAct (as in (Yao et al., 2022b)) and our variations along RQ1, RQ2 and RQ3 for the `Put` task.

### D.1   Synonym Substitution mapping for `Domain`

We make the following substitutions to the object names / locations in the exemplar prompt in the `Domain` variation. Note that these substitutions are done only to the exemplar, and the query problem and subsequent interaction with the AlfWorld simulator uses the original vocabulary mapping.

```
spraybottle -> aerosolbottle
cabinet -> cupboard
countertop -> worktop
sinkbasin -> sinkbowl
toilet -> lavatory
toiletpaperhanger -> toiletpaperholder
towelholder -> towelrack

microwave -> oven
shelf -> rack
drawer -> deskdrawer
stoveburner -> hob
diningtable -> table
garbagecan -> trashbin

fridge -> refrigerator
peppershaker -> pepperpot
room -> livingroom
bread -> breadloaf
pan -> fryingpan
pot -> saucepan
book -> notebook

creditcard -> amexcard
mirror -> lookingglass
dresser -> chestofdrawers
sofa -> couch
cellphone -> mobilephone
coffeemachine -> coffeemaker
knife -> kitchenknife
spatula -> turner
soapbottle -> liquidsoapdispenser
saltshaker -> saltpot
statue -> sculpture
vase -> flowerpot
dishsponge -> spongewipe
```

```
desklamp -> tablelamp
sidetable -> nightstand
```

## D.2 For `All, Both, One`

**All**: We take the exemplar prompt for each task and concatenate it together.

**Both** : We use the following mapping generated arbitrarily, to replace the exemplar prompt for the query task.

```
VARIATION_MAPPING = {
    'put':'examine',
    'clean':'cool',
    'heat':'put',
    'cool':'puttwo',
    'examine':'clean',
    'puttwo':'heat'
}
```

**One** : We use the first example from the above mapping, and the second example from the same task as the query.

## D.3 Building the input prompt

We inherit ReAct's way of stiching together the exemplars and the query using the following prompt template as :

```
Interact with a household to solve a task.  Here are two examples.'  + eg1 + eg2 +
INSTRUCTION + Here is the task.  (TASK)´
```

In the case of GPT models `INSTRUCTION` is an empty string. Claude Models gave 0% success rate (regardless of the model size, i.e. Haiku, Sonnet and Opus) with the original ReAct prompt design. We identified that Claude required an explicit instruction to generate only the actions for the simulator. We added a task independent INSTRUCTION prompt as :

```
INSTRUCTION = " Only provide the one action at a time.  Be concise, and do not provide
any extra information.  Always start with the action.  For example, 'pick up apple' is
correct, but 'I want to pick up the apple' is not."
```

Interestingly, GPT-3-Turbo performance dropped to 1% (averaged across 6 independent runs over 134 examples) with the inclusion of this INSTRUCTION tag. This further highlights how brittle prompt engineering is, and cautions us against misplacing our trust with ReAct in enhancing reasoning abilities of LLMs.

Note that, we did experiment with both prompt variations (with our explicit instruction and without) for Claude models and picked the best-performing variant (which had the instructions) for our reported results.

## D.4 RQ1

---

**Prompt 1: rq1-anon-exemplar**

You are in the middle of a room. Looking quickly around you, you see a cabinet 4, a cabinet 3, a cabinet 2, a cabinet 1, a coffeemachine 1, a countertop 1, a diningtable 3, a diningtable 2, a diningtable 1, a drawer 1, a fridge 1, a garbagecan 1, a microwave 1, a sidetable 1, a sinkbasin 1, a stoveburner 4, a stoveburner 3, a stoveburner 2, a stoveburner 1, and a toaster 1.
Your task is to: find some apple and put it in sidetable.
> think: To solve the task, I need to find and take the object, then put it on the desired location. First I need to find the object. The object is more likely to appear in one of the different locations. I can check one by one, starting with the first location. Once I find the object, next I need to take it. Once I take the object, next I need to put it in/on the desired location.
OK.
> go to fridge 1
The fridge 1 is closed.
> open fridge 1
You open the fridge 1. The fridge 1 is open. In it, you see a lettuce 2, a mug 2, a potato 2, and a tomato 1.
> go to diningtable 1
On the diningtable 1, you see nothing.
> go to diningtable 2
On the diningtable 2, you see a pineapple 1.
> go to diningtable 3
On the diningtable 3, you see a bowl 2, a bowl 1, a bread 2, and a bread 1.
> go to sidetable 1
On the sidetable 1, you see a cup 1, a lettuce 1, a peppershaker 3, a potato 1, and a saltshaker 1.
> go to countertop 1
On the countertop 1, you see a butterknife 1, a dishsponge 1, and a soapbottle 1.
> go to sinkbasin 1
On the sinkbasin 1, you see a cup 2, a egg 1, a mug 1, and a potato 3.
> go to garbagecan 1
On the garbagecan 1, you see a apple 3, and a egg 3.
> take apple 3 from garbagecan 1
You pick up the apple 3 from the garbagecan 1.
> go to sidetable 1
On the sidetable 1, you see a cup 1, a lettuce 1, a peppershaker 3, a potato 1, and a saltshaker 1.
> put apple 3 in/on sidetable 1
You put the apple 3 in/on the sidetable 1.

---

**Prompt 2: rq1-exemplar**

You are in the middle of a room. Looking quickly around you, you see a cabinet 4, a cabinet 3, a cabinet 2, a cabinet 1, a coffeemachine 1, a countertop 1, a diningtable 3, a diningtable 2, a diningtable 1, a drawer 1, a fridge 1, a garbagecan 1, a microwave 1, a sidetable 1, a sinkbasin 1, a stoveburner 4, a stoveburner 3, a stoveburner 2, a stoveburner 1, and a toaster 1.
Your task is to: find some apple and put it in sidetable.
> think: To solve the task, I need to find and take an apple, then put it in sidetable. First I need to find an apple. An apple is more likely to appear in fridges (1), diningtables (1-3), sidetables (1), countertops (1), sinkbasins (1), garbagecan (1). I can check one by one, starting with fridge 1. Once I find an apple, next I need to take it. Once I find an apple, next I need to put it in/on sidetable.
OK.
> go to fridge 1
The fridge 1 is closed.
> open fridge 1
You open the fridge 1. The fridge 1 is open. In it, you see a lettuce 2, a mug 2, a potato 2, and a tomato 1.
> go to diningtable 1
On the diningtable 1, you see nothing.
> go to diningtable 2
On the diningtable 2, you see a pineapple 1.
> go to diningtable 3
On the diningtable 3, you see a bowl 2, a bowl 1, a bread 2, and a bread 1.
> go to sidetable 1
On the sidetable 1, you see a cup 1, a lettuce 1, a peppershaker 3, a potato 1, and a saltshaker 1.
> go to countertop 1
On the countertop 1, you see a butterknife 1, a dishsponge 1, and a soapbottle 1.
> go to sinkbasin 1
On the sinkbasin 1, you see a cup 2, a egg 1, a mug 1, and a potato 3.
> go to garbagecan 1
On the garbagecan 1, you see a apple 3, and a egg 3.
> take apple 3 from garbagecan 1
You pick up the apple 3 from the garbagecan 1.
> go to sidetable 1
On the sidetable 1, you see a cup 1, a lettuce 1, a peppershaker 3, a potato 1, and a saltshaker 1.
> put apple 3 in/on sidetable 1
You put the apple 3 in/on the sidetable 1.

---

---

**Prompt 3: base-react**

You are in the middle of a room. Looking quickly around you, you see a cabinet 4, a cabinet 3, a cabinet 2, a cabinet 1, a coffeemachine 1, a countertop 1, a diningtable 3, a diningtable 2, a diningtable 1, a drawer 1, a fridge 1, a garbagecan 1, a microwave 1, a sidetable 1, a sinkbasin 1, a stoveburner 4, a stoveburner 3, a stoveburner 2, a stoveburner 1, and a toaster 1.
Your task is to: find some apple and put it in sidetable.
> think: First I need to find an apple. An apple is more likely to appear in fridges (1), diningtables (1-3), sidetables (1), countertops (1), sinkbasins (1), garbagecan (1). I can check one by one, starting with fridge 1.
OK.
> go to fridge 1
The fridge 1 is closed.
> open fridge 1
You open the fridge 1. The fridge 1 is open. In it, you see a lettuce 2, a mug 2, a potato 2, and a tomato 1.
> go to diningtable 1
On the diningtable 1, you see nothing.
> go to diningtable 2
On the diningtable 2, you see a pineapple 1.
> go to diningtable 3
On the diningtable 3, you see a bowl 2, a bowl 1, a bread 2, and a bread 1.
> go to sidetable 1
On the sidetable 1, you see a cup 1, a lettuce 1, a peppershaker 3, a potato 1, and a saltshaker 1.
> go to countertop 1
On the countertop 1, you see a butterknife 1, a dishsponge 1, and a soapbottle 1.
> go to sinkbasin 1
On the sinkbasin 1, you see a cup 2, a egg 1, a mug 1, and a potato 3.
> go to garbagecan 1
On the garbagecan 1, you see a apple 3, and a egg 3.
> take apple 3 from garbagecan 1
You pick up the apple 3 from the garbagecan 1.
> think: Now I find a apple (3). Next, I need to put it in/on sidetable 1.
OK.
> go to sidetable 1
On the sidetable 1, you see a cup 1, a lettuce 1, a peppershaker 3, a potato 1, and a saltshaker 1.
> put apple 3 in/on sidetable 1
You put the apple 3 in/on the sidetable 1.

### D.5  RQ2

---

**Prompt 4: rq2-Explanation**

You are in the middle of a room. Looking quickly around you, you see a cabinet 4, a cabinet 3, a cabinet 2, a cabinet 1, a coffeemachine 1, a countertop 1, a diningtable 3, a diningtable 2, a diningtable 1, a drawer 1, a fridge 1, a garbagecan 1, a microwave 1, a sidetable 1, a sinkbasin 1, a stoveburner 4, a stoveburner 3, a stoveburner 2, a stoveburner 1, and a toaster 1.
Your task is to: find some apple and put it in sidetable.
> go to fridge 1
The fridge 1 is closed.
> close cabinet 2.
Nothing Happens.
> think: Nothing happens because you are not at cabinet 2.
OK.
> open fridge 1
You open the fridge 1. The fridge 1 is open. In it, you see a lettuce 2, a mug 2, a potato 2, and a tomato 1.
> go to diningtable 1
On the diningtable 1, you see nothing.
> go to diningtable 2
On the diningtable 2, you see a pineapple 1.
> go to diningtable 3
On the diningtable 3, you see a bowl 2, a bowl 1, a bread 2, and a bread 1.
> go to sidetable 1
On the sidetable 1, you see a cup 1, a lettuce 1, a peppershaker 3, a potato 1, and a saltshaker 1.
> go to countertop 1
On the countertop 1, you see a butterknife 1, a dishsponge 1, and a soapbottle 1.
> go to sinkbasin 1
On the sinkbasin 1, you see a cup 2, a egg 1, a mug 1, and a potato 3.
> go to garbagecan 1
On the garbagecan 1, you see a apple 3, and a egg 3.
> put apple 3 in/on sidetable.
Nothing Happens.
> think: Nothing happens because I do not have apple 3.
OK.
> take apple 3 from garbagecan 1
You pick up the apple 3 from the garbagecan 1.
> go to sidetable 1
On the sidetable 1, you see a cup 1, a lettuce 1, a peppershaker 3, a potato 1, and a saltshaker 1.
> put apple 3 in/on sidetable 1
You put the apple 3 in/on the sidetable 1.

---

**Prompt 5: rq2-ordering**

You are in the middle of a room. Looking quickly around you, you see a cabinet 4, a cabinet 3, a cabinet 2, a cabinet 1, a coffeemachine 1, a countertop 1, a diningtable 3, a diningtable 2, a diningtable 1, a drawer 1, a fridge 1, a garbagecan 1, a microwave 1, a sidetable 1, a sinkbasin 1, a stoveburner 4, a stoveburner 3, a stoveburner 2, a stoveburner 1, and a toaster 1.
Your task is to: find some apple and put it in sidetable.
> think: To solve the task, I need to put an apple in sidetable, for which I need to find and take an apple.
OK.
> think: An apple is more likely to appear in sidetables (1), fridges (1), countertops (1), diningtables (1-3), garbagecan (1), sinkbasins (1). First I need to find an apple. I can check one by one, starting with fridge 1.
OK.
> go to fridge 1
The fridge 1 is closed.
> open fridge 1
You open the fridge 1. The fridge 1 is open. In it, you see a lettuce 2, a mug 2, a potato 2, and a tomato 1.
> go to diningtable 1
On the diningtable 1, you see nothing.
> go to diningtable 2
On the diningtable 2, you see a pineapple 1.
> go to diningtable 3
On the diningtable 3, you see a bowl 2, a bowl 1, a bread 2, and a bread 1.
> go to sidetable 1
On the sidetable 1, you see a cup 1, a lettuce 1, a peppershaker 3, a potato 1, and a saltshaker 1.
> go to countertop 1
On the countertop 1, you see a butterknife 1, a dishsponge 1, and a soapbottle 1.
> go to sinkbasin 1
On the sinkbasin 1, you see a cup 2, a egg 1, a mug 1, and a potato 3.
> go to garbagecan 1
On the garbagecan 1, you see a apple 3, and a egg 3.
> think: Next, I need to take the apple.Now I find an apple (3).
OK.
> take apple 3 from garbagecan 1
You pick up the apple 3 from the garbagecan 1.
> think: Next, I need to put the apple in/on sidetable 1. Now I find an apple (3).
OK.
> go to sidetable 1
On the sidetable 1, you see a cup 1, a lettuce 1, a peppershaker 3, a potato 1, and a saltshaker 1.
> put apple 3 in/on sidetable 1
You put the apple 3 in/on the sidetable 1.

---

**Prompt 6: rq2-Placebo**

You are in the middle of a room. Looking quickly around you, you see a cabinet 4, a cabinet 3, a cabinet 2, a cabinet 1, a coffeemachine 1, a countertop 1, a diningtable 3, a diningtable 2, a diningtable 1, a drawer 1, a fridge 1, a garbagecan 1, a microwave 1, a sidetable 1, a sinkbasin 1, a stoveburner 4, a stoveburner 3, a stoveburner 2, a stoveburner 1, and a toaster 1.
Your task is to: find some apple and put it in sidetable.
> think: Take a deep breath and work on this problem step-by-step.
OK.
> go to fridge 1
The fridge 1 is closed.
> open fridge 1
You open the fridge 1. The fridge 1 is open. In it, you see a lettuce 2, a mug 2, a potato 2, and a tomato 1.
> go to diningtable 1
On the diningtable 1, you see nothing.
> go to diningtable 2
On the diningtable 2, you see a pineapple 1.
> go to diningtable 3
On the diningtable 3, you see a bowl 2, a bowl 1, a bread 2, and a bread 1.
> go to sidetable 1
On the sidetable 1, you see a cup 1, a lettuce 1, a peppershaker 3, a potato 1, and a saltshaker 1.
> go to countertop 1
On the countertop 1, you see a butterknife 1, a dishsponge 1, and a soapbottle 1.
> go to sinkbasin 1
On the sinkbasin 1, you see a cup 2, a egg 1, a mug 1, and a potato 3.
> go to garbagecan 1
On the garbagecan 1, you see a apple 3, and a egg 3.
> think: Take a deep breath and work on this problem step-by-step.
OK.
> take apple 3 from garbagecan 1
You pick up the apple 3 from the garbagecan 1.
> think: Take a deep breath and work on this problem step-by-step.
OK.
> go to sidetable 1
On the sidetable 1, you see a cup 1, a lettuce 1, a peppershaker 3, a potato 1, and a saltshaker 1.
> put apple 3 in/on sidetable 1
You put the apple 3 in/on the sidetable 1.

---

**Prompt 7: rq2-Failure**

You are in the middle of a room. Looking quickly around you, you see a cabinet 4, a cabinet 3, a cabinet 2, a cabinet 1, a coffeemachine 1, a countertop 1, a diningtable 3, a diningtable 2, a diningtable 1, a drawer 1, a fridge 1, a garbagecan 1, a microwave 1, a sidetable 1, a sinkbasin 1, a stoveburner 4, a stoveburner 3, a stoveburner 2, a stoveburner 1, and a toaster 1.
Your task is to: find some apple and put it in sidetable.
> go to fridge 1
The fridge 1 is closed.
> close cabinet 2.
Nothing Happens.
> open fridge 1
You open the fridge 1. The fridge 1 is open. In it, you see a lettuce 2, a mug 2, a potato 2, and a tomato 1.
> go to diningtable 1
On the diningtable 1, you see nothing.
> go to diningtable 2
On the diningtable 2, you see a pineapple 1.
> go to diningtable 3
On the diningtable 3, you see a bowl 2, a bowl 1, a bread 2, and a bread 1.
> go to sidetable 1
On the sidetable 1, you see a cup 1, a lettuce 1, a peppershaker 3, a potato 1, and a saltshaker 1.
> go to countertop 1
On the countertop 1, you see a butterknife 1, a dishsponge 1, and a soapbottle 1.
> go to sinkbasin 1
On the sinkbasin 1, you see a cup 2, a egg 1, a mug 1, and a potato 3.
> go to garbagecan 1
On the garbagecan 1, you see a apple 3, and a egg 3.
> put apple 3 in/on sidetable.
Nothing Happens.
> take apple 3 from garbagecan 1
You pick up the apple 3 from the garbagecan 1.
> go to sidetable 1
On the sidetable 1, you see a cup 1, a lettuce 1, a peppershaker 3, a potato 1, and a saltshaker 1.
> put apple 3 in/on sidetable 1
You put the apple 3 in/on the sidetable 1.

**D.6    RQ3**

---

**Prompt 8: rq3-Domain**

You are in the middle of a livingroom. Looking quickly around you, you see a cupboard 4, a cupboard 3, a cupboard 2, a cupboard 1, a coffeemaker 1, a worktop 1, a table 3, a table 2, a table 1, a deskdrawer 1, a refrigerator 1, a trashbin 1, a oven 1, a nightstand 1, a sinkbowl 1, a hob 4, a hob 3, a hob 2, a hob 1, and a toaster 1.
Your task is to: find some apple and put it in nightstand.
> think: To solve the task, I need to find and take an apple, then put it in nightstand.
OK.
> think: First I need to find an apple. An apple is more likely to appear in refrigerators (1), tables (1-3), nightstands (1), worktops (1), sinkbowls (1), trashbin (1). I can check one by one, starting with refrigerator 1.
OK.
> go to refrigerator 1
The refrigerator 1 is closed.
> oballpointpen refrigerator 1
You oballpointpen the refrigerator 1. The refrigerator 1 is oballpointpen. In it, you see a lettuce 2, a mug 2, a saucepanato 2, and a tomato 1.
> go to table 1
On the table 1, you see nothing.
> go to table 2
On the table 2, you see a pineapple 1.
> go to table 3
On the table 3, you see a bowl 2, a bowl 1, a breadloaf 2, and a breadloaf 1.
> go to nightstand 1
On the nightstand 1, you see a cup 1, a lettuce 1, a peppersaucepan 3, a saucepanato 1, and a saltpot 1.
> go to worktop 1
On the worktop 1, you see a butterknife 1, a spongewipe 1, and a liquidsoapdispenser 1.
> go to sinkbowl 1
On the sinkbowl 1, you see a cup 2, a egg 1, a mug 1, and a saucepanato 3.
> go to trashbin 1
On the trashbin 1, you see a apple 3, and a egg 3.
> think: Now I find an apple (3). Next, I need to take it.
> take apple 3 from trashbin 1
You pick up the apple 3 from the trashbin 1.
> think: Now I find an apple (3). Next, I need to put it in/on nightstand 1.
OK.
> go to nightstand 1
On the nightstand 1, you see a cup 1, a lettuce 1, a peppersaucepan 3, a saucepanato 1, and a saltpot 1.
> put apple 3 in/on nightstand 1
You put the apple 3 in/on the nightstand 1.

---

---

**Prompt 9: rq3-Instance**

You are in the middle of a room. Looking quickly around you, you see a cabinet 4, a cabinet 3, a cabinet 2, a cabinet 1, a coffeemachine 1, a countertop 1, a diningtable 3, a diningtable 2, a diningtable 1, a drawer 1, a fridge 1, a garbagecan 1, a microwave 1, a sidetable 1, a sinkbasin 1, a stoveburner 4, a stoveburner 3, a stoveburner 2, a stoveburner 1, and a toaster 1.
Your task is to: find some apple and put it in sidetable.
> think: To solve the task, I need to find and take an apple, then put it in sidetable.
OK.
> think: First I need to find an apple. An apple is more likely to appear in fridges (1), diningtables (1-3), sidetables (1), countertops (1), sinkbasins (1), garbagecan (1). I can check one by one, starting with fridge 1.
OK.
> go to fridge 1
The fridge 1 is closed.
> open fridge 1
You open the fridge 1. The fridge 1 is open. In it, you see a lettuce 2, a mug 2, a potato 2, and a tomato 1.
> go to diningtable 1
On the diningtable 1, you see nothing.
> go to diningtable 2
On the diningtable 2, you see a pineapple 1.
> go to diningtable 3
On the diningtable 3, you see a bowl 2, a bowl 1, a bread 2, and a bread 1.
> go to sidetable 1
On the sidetable 1, you see a cup 1, a lettuce 1, a peppershaker 3, a potato 1, and a saltshaker 1.
> go to countertop 1
On the countertop 1, you see a butterknife 1, a dishsponge 1, and a soapbottle 1.
> go to sinkbasin 1
On the sinkbasin 1, you see a cup 2, a egg 1, a mug 1, and a potato 3.
> go to diningtable 2
On the diningtable 2, you see a pineapple 1.
> go to diningtable 3
On the diningtable 3, you see a bowl 2, a bowl 1, a bread 2, and a bread 1.
> go to sidetable 1
On the sidetable 1, you see a cup 1, a lettuce 1, a peppershaker 3, a potato 1, and a saltshaker 1.
> go to countertop 1
On the countertop 1, you see a butterknife 1, a dishsponge 1, and a soapbottle 1.
> go to sinkbasin 1
On the sinkbasin 1, you see a cup 2, a egg 1, a mug 1, and a potato 3.
> go to garbagecan 1
On the garbagecan 1, you see a apple 3, and a egg 3.
> think: Now I find an apple (3). Next, I need to take it.
> take apple 3 from garbagecan 1
You pick up the apple 3 from the garbagecan 1.
> think: Now I find an apple (3). Next, I need to put it in/on sidetable 1.
OK.
> go to sidetable 1
On the sidetable 1, you see a cup 1, a lettuce 1, a peppershaker 3, a potato 1, and a saltshaker 1.
> put apple 3 in/on sidetable 1
You put the apple 3 in/on the sidetable 1.

---

**Prompt 10: rq3-Optimal**

You are in the middle of a room. Looking quickly around you, you see a cabinet 4, a cabinet 3, a cabinet 2, a cabinet 1, a coffeemachine 1, a countertop 1, a diningtable 3, a diningtable 2, a diningtable 1, a drawer 1, a fridge 1, a garbagecan 1, a microwave 1, a sidetable 1, a sinkbasin 1, a stoveburner 4, a stoveburner 3, a stoveburner 2, a stoveburner 1, and a toaster 1.
Your task is to: find some apple and put it in sidetable.
> think: To solve the task, I need to find and take an apple, then put it in sidetable.
OK.
> think: First I need to find an apple. An apple is more likely to appear in fridges (1), diningtables (1-3), sidetables (1), countertops (1), sinkbasins (1), garbagecan (1). I can check one by one, starting with garbagecan 1.
OK.
> go to garbagecan 1
On the garbagecan 1, you see a apple 3, and a egg 3.
> think: Now I find an apple (3). Next, I need to take it.
> take apple 3 from garbagecan 1
You pick up the apple 3 from the garbagecan 1.
> think: Now I find an apple (3). Next, I need to put it in/on sidetable 1.
OK.
> go to sidetable 1
On the sidetable 1, you see a cup 1, a lettuce 1, a peppershaker 3, a potato 1, and a saltshaker 1.
> put apple 3 in/on sidetable 1
You put the apple 3 in/on the sidetable 1.

