# OpenReview forum: "Do Think Tags Really Help LLMs Plan?  A Critical Evaluation of ReAct-Style Prompting"
_TMLR — Accepted by TMLR_

### Review · Reviewer_MVGU · 2025-03-27

**Summary Of Contributions:**

This paper critically evaluates the effectiveness of ReAct-style prompting, specifically its claim that interleaving reasoning traces ("think" tags) with action execution improves the planning abilities of Large Language Models (LLMs). The authors conduct a sensitivity analysis by systematically varying aspects of the ReAct prompt in the AlfWorld and WebShop domains, using various LLMs including GPT-3.5, GPT-4, GPT-4o, Claude, and Llama 3.1-8B.

The key contributions and findings are:

1.  **Interleaving reasoning is not the primary driver of performance:** Contrary to ReAct's claims, the study finds that LLM performance often *improves* when the reasoning trace is *not* interleaved with actions (RQ1). Chain-of-Thought (CoT) style prompting, where reasoning is done upfront, generally performs better or comparably to ReAct.
2.  **Content of reasoning traces has minimal impact:** The specific content or structure of the "think" tags (e.g., providing foresight guidance vs. hindsight/failure explanation vs. placebo text) has a surprisingly small effect on performance (RQ2). Even placebo guidance performs comparably in many cases, suggesting the reasoning content itself might not be crucial.
3.  **High dependence on exemplar-query similarity:** The study demonstrates a severe lack of generalization. Performance drastically drops when there are even minor variations between the few-shot examples in the prompt and the queried task (RQ3). This includes using synonyms for objects/locations, changing the goal instance slightly, or using examples from different but related tasks within the same domain. The LLMs appear to rely heavily on literal pattern matching or approximate retrieval from the provided examples rather than genuine reasoning or planning.
4.  **Questioning emergent reasoning:** The results challenge the notion that ReAct successfully elicits robust reasoning and planning abilities. The observed performance seems largely attributable to the high similarity between prompts and queries, forcing prompt engineers to provide highly instance-specific examples, which limits scalability and practical applicability.
5.  **Reproducibility and Consistency Issues:** The authors note discrepancies with original ReAct results (which used PaLM) when using newer models, and highlight inconsistencies in performance even across different versions of the same model family (e.g., GPT-4 variants). Performance of the simple "Act" baseline (no think tags) is often surprisingly strong, further questioning the added value of ReAct's core mechanism.

In essence, the paper argues that the perceived benefits of ReAct prompting stem primarily from the close similarity between few-shot examples and the target task (acting like a nearest-neighbor retrieval mechanism) rather than the interleaved reasoning traces themselves. This "brittleness" and lack of generalization call into question the claims of enhanced reasoning and planning capabilities fostered by the ReAct method.

**Audience:**

Yes

**Broader Impact Concerns:**

The paper primarily focuses on a technical evaluation of a specific LLM prompting technique (ReAct) and its claimed effectiveness for reasoning and planning tasks. The core findings relate to the limitations of this technique and the nature of LLM performance (pattern matching vs. reasoning).

Based on this focus, there do not appear to be immediate, direct ethical implications that would necessitate a dedicated Broader Impact Statement beyond what is standard for ML research. The work does not:

*   Introduce new datasets containing sensitive information.
*   Deploy models in high-stakes real-world scenarios.
*   Directly address fairness, bias, or malicious use cases in a way that introduces novel risks.

However, one could consider **indirect implications**:

1.  **Over-reliance on LLMs for Critical Tasks:** By highlighting the brittleness and lack of true generalization/reasoning in current LLM prompting methods (even sophisticated ones like ReAct), the work implicitly cautions against deploying these systems in critical decision-making or planning scenarios where robustness and true understanding are required. This finding itself promotes responsible AI development by revealing limitations.
2.  **Cost and Environmental Impact:** The experiments involve significant computation using large models (GPT-4, Claude Opus). While the authors provide token estimates (Appendix A), acknowledging the environmental and financial costs associated with such large-scale LLM experimentation is generally good practice in the field, though not typically requiring a full BIS unless the scale is exceptionally massive or introduces novel efficiency concerns.

These points are general considerations for much LLM research rather than specific ethical risks raised uniquely by this paper's methodology or findings. Therefore, a separate Broader Impact Statement is likely not strictly necessary, although a brief mention of the implications for responsible deployment given the observed lack of robustness could be considered for the conclusion or discussion.

**Claims And Evidence:**

Yes

**Requested Changes:**

Here are proposed adjustments, categorized by importance:

**Critical Adjustments:**

1.  **None.** The paper's core methodology, results, and conclusions are sound and well-supported by the extensive experiments. The findings convincingly challenge the claims of ReAct based on the provided evidence.

**Suggested Adjustments (to strengthen the work):**

1.  **Elevate Discussion of "Act" Baseline Performance:** In Sections 5.1, 5.2, and the Discussion (Sec 6), more explicitly highlight and discuss the implications of the "Act" baseline often outperforming ReAct, especially with newer models (Tables 1 & 2). This directly contradicts ReAct's core premise and strengthens the paper's argument significantly. *[Strength: High - directly supports the main thesis]*
2.  **Refine Interpretation of WebShop Results:** Acknowledge the extremely low absolute success rates in WebShop more directly in Section 5.3. Discuss the potential limitations this imposes on interpreting the *relative* differences between prompting strategies in this domain. Briefly speculate on *why* performance might be so degraded compared to the original ReAct paper (beyond just random sampling, e.g., model changes, API differences?). *[Strength: Medium - adds nuance and addresses potential reviewer concerns]*
3.  **Expand Analysis of Generated Thoughts (Optional but valuable):** If feasible, briefly expand Section 6 ('Operationalizing ‘think’ actions') or add an appendix section with qualitative examples or quantitative analysis (beyond just invalid action rates) of the *content* generated within `think` tags by the LLMs under different conditions (e.g., ReAct vs. CoT vs. Placebo). Does the LLM generate meaningful reasoning, or just boilerplate text? This could provide deeper insight into the mechanism (or lack thereof). *[Strength: Medium - provides deeper mechanistic insight]*
4.  **Quantify "Similarity" (Optional):** Consider adding a brief discussion or supplementary analysis exploring more quantitative measures of exemplar-query similarity (e.g., BLEU score, ROUGE, embedding cosine similarity between task descriptions or full traces) for the RQ3 variations (especially `Domain`, `One`, `Both`). Correlating these quantitative similarities with the observed performance drops could add another layer of evidence to the claim that literal similarity is the key driver. *[Strength: Low-Medium - adds rigor but may be complex to implement thoroughly]*
5.  **Clarify Hyperparameters/Implementation Details:**
    *   Ensure the referenced supplementary material/code is accessible and contains the full prompts and implementation details as stated (Appendix C).
    *   Briefly state the number of examples used for each variation (e.g., "two exemplars were used for all variations except 'All' which used six"). This is implied but could be stated explicitly for clarity early in Section 4 or C. *[Strength: Low - improves clarity and reproducibility]*

**Strengths And Weaknesses:**

**Strong Aspects:**

1.  **Timely and Relevant Topic:** The paper addresses a highly relevant and debated topic: the actual reasoning and planning capabilities of LLMs, specifically focusing on the popular ReAct prompting method. Given the widespread adoption of ReAct (cited >2400 times), a critical evaluation is valuable.
2.  **Systematic and Rigorous Evaluation:** The authors conduct a comprehensive sensitivity analysis, systematically isolating and testing the core claims of ReAct. They design specific variations targeting the interleaving aspect (RQ1), the content of reasoning (RQ2), and the exemplar-query similarity (RQ3). This methodical approach is a significant strength.
3.  **Clear Research Questions:** The study is well-structured around three clear research questions (RQ1, RQ2, RQ3), making the investigation focused and easy to follow.
4.  **Reproducibility Efforts:** The authors test their hypotheses across multiple modern LLMs (various GPT versions, Claude, Llama) and in the original domains used by ReAct (AlfWorld, WebShop), strengthening the generalizability of their findings and addressing potential model-specific artefacts. They also provide details on resources used and experiment design, including links to code (though availability wasn't verified).
5.  **Compelling Results:** The findings consistently challenge the core tenets of ReAct across different models and domains. The strong dependence on exemplar-query similarity (RQ3) and the relatively minor impact of reasoning content/interleaving (RQ1, RQ2) provide compelling evidence against ReAct's claimed mechanism of action.
6.  **Contextualization with Existing Literature:** The paper effectively positions its findings within the broader context of research on LLM reasoning, prompt engineering, and critiques of emergent abilities, citing relevant contemporary work.
7.  **Clear Presentation:** The paper is generally well-written and clearly presents its methodology, results (using tables and radar charts effectively), and conclusions. The figures illustrating prompt variations (Fig 2) are helpful.

**Weak Aspects / Requiring Attention:**

1.  **"Act" Baseline Definition:** While the "Act" baseline is mentioned (no "think" tags), its performance relative to ReAct and other variations (especially in Tables 1 & 2) is crucial but sometimes understated in the text. The fact that "Act" often performs *better* than ReAct on newer models is a key finding that could be emphasized more as direct evidence against ReAct's utility.
2.  **WebShop Results Interpretation:** The performance on WebShop is very low across *all* methods (single digits in Table 2). The authors attribute this to decoupling exemplars and queries due to random sampling. While plausible, this extremely low baseline makes it harder to draw strong conclusions about the *relative* differences between methods in this domain. The discussion could elaborate on why performance is so low and the implications for evaluating prompting methods when task success itself is rare.
3.  **Analysis of Generated "Think" Content:** While RQ2 varies the *input* "think" content, the paper only briefly touches upon the *quality* or *validity* of the "think" content generated by the LLM itself during inference (Section 6). A deeper analysis of *what* the LLM generates when prompted with different variations (especially ReAct vs. CoT) could offer further insights into *why* performance varies. The observation that 40%-90% of subsequent actions after a `think` tag are invalid is interesting and could be explored further.
4.  **Definition of "Similarity":** The concept of "exemplar-query similarity" is central to the paper's main conclusion (RQ3). While the variations (Domain, Instance, One, Both) operationalize different facets of similarity, the term itself remains somewhat qualitative. Could more quantitative measures of similarity (e.g., embedding distance, lexical overlap) be used to correlate with performance drops? This might further solidify the link between similarity and performance.
5.  **Optimal Prompt Placement (Minor):** Appendix C.1 mentions testing different placements for the query-task exemplar in the `All` variation. While it's stated the end position was best, including this data point, even briefly in the appendix, might add completeness.

---

> ### Author Response · Authors · 2025-04-16
> **Rebuttal**
>
> We would like to thank the reviewer for their thoughtful comments and detailed feedback. We respond to the concerns raised below:
>
> 1. $\textbf{``Act" Baseline Definition:}$ We have included a more detailed description of the Act baseline along with an example for a better understanding of the readers to make a uniform comparison with the ReAct framework in the beginning of Section 5 (pg 7). As pointed out, we have further emphasized the results where the Act baseline outperforms the ReAct framework across multiple LLMs in the Discussion section as well (pg 12).
>
> 2. $\textbf{WebShop Results Interpretation:}$ We have run additional tests with WebShop following a similar sampling strategy of 500 test problems as has been done in ReAct and for the current set of experiments in the paper with GPT-4o. We notice an exact similar pattern in performance as seen earlier with Act-only baseline and other prompt variations performing better than the ReAct framework. We have reported these results in Appendix Section B.5 (page 19).
> One possible reason for lower accuracy numbers across the variations may possibly be because of the model differences assuming that the PaLM model (originally used by ReAct, but currently decommissioned) was particularly suited for WebShop unlike the newer models that we have tested.
>
> 3. $\textbf{Analysis of Generated ``think" Content:}$ In the revised PDF’s appendix in Section B.4 (page 17 onwards), we have included a new subsection of an example of the ‘think’ tag content for the AlfWorld domain  across the different prompt variation settings shown in the work. With the help of these examples, we have tried showing how LLMs try to replicate the content given in the ‘think’ tag of the in-context demonstrations/examples, and where they generate invalid content for these ‘think’ tags.
>
> 4. $\textbf{Definition of ``Similarity":}$ We did a preliminary study on the similarity (computed in terms of embedding distances) between the 134 prompting tasks with the originally provided examples, and the performance of ReACT. As can be seen from the figures we have added in Appendix Section B.6 (page 20), as expected, the performance degrades significantly in the off-diagonal cases. That said, we should point out that embedding distances and lexical distances are considered poor surrogates for assessing task similarity and goal similarity. In the automated planning community, they use more semantic notions [1], which is dependent on the knowledge of the domain model.
>
> 5. $\textbf{Clarify Hyperparameters/Implementation Details:}$ We have ensured that the supplementary contains the zipped code and prompts for ease of reproducibility. We have also added the detail on the number of in-context examples used for RQ3 in Section 4 when describing the different variations.
>
> 6. $\textbf{Broader Impact statement:}$  We have included a Broader Impact statement at the end of our work discussing the two points suggested by the reviewer on the over-reliance on LLMs especially for critical applications, and other considerations that come along with using these models.
>
> [1] Kulkarni, Anagha, et al. "Explicable planning as minimizing distance from expected behavior." AAMAS Conference proceedings. 2019.

---

### Review · Reviewer_tvtd · 2025-04-01

**Summary Of Contributions:**

This paper explores the impact of different prompt variations on the claims made in ReAct. The authors design three types of modifications:

- Altering the format of interleaved thinking and acting, such as adopting a CoT-style approach.

- Modifying reasoning traces, e.g., introducing invalid actions or reversing the guidance order.

- Adjusting the similarity between exemplars and queries, such as using synonyms or selecting exemplars from different tasks.

Experiments are conducted using various GPT models, Claude-Opus, and LLAMA-3.1-8B to evaluate Act, ReAct, and the proposed variations. The findings suggest that interleaving thinking and acting, as well as the context of reasoning traces, have minimal impact on performance. Instead, exemplar-query similarity is the primary influencing factor.

**Audience:**

Yes

**Broader Impact Concerns:**

None.

**Claims And Evidence:**

Yes

**Requested Changes:**

Simply strengthen the work in my view: Variation 4 - Exploration Strategy in Section 4.3 is hard to understand.

**Strengths And Weaknesses:**

Strengths:
- The paper carefully designs prompt variations to examine ReAct’s capabilities.
- A diverse set of models is evaluated using these variations, leading to insightful conclusions about potential factors influencing ReAct’s performance.

Weaknesses:
- While prompt design is a key factor, performance is influenced by multiple aspects, including model choice and evaluation benchmarks.
- The experimental results support the claims within the given setup, but they may not be sufficient to fully justify ReAct’s effectiveness, especially in complex, real-world tasks beyond benchmarks like AlfWorld and WebShop.
- The finding that exemplar-query similarity plays a crucial role is not surprising, as it aligns with common knowledge in the community.

---

> ### Author Response · Authors · 2025-04-16
> **Rebuttal**
>
> We would like to thank the reviewer for their thoughtful comments and detailed feedback. We respond to the concerns raised below:
>
> 1. $\textbf{Choice of evaluation benchmarks:}$ We would like to highlight that AlfWorld and WebShop are two of the most popular text based sequential decision-making tasks that have elements of real world interaction being used by various contemporary research works accepted at top tier venues [1, 2, 3, 4]. Furthermore, as pointed out in the paper, our attempt has been to perform a deep and pointed analysis of the brittleness of ReAct rather than support our analysis via broad and shallow empirical results. We believe that it is the broad and shallow approach to experimental analysis of large language model abilities that led to the popularity of brittle prompt engineering methods. Similar cases have been found with the planning and reasoning abilities of large language models where newer works with focused experimental analysis have now pinpointed the shortcomings of the reasoning abilities of large language models [5, 6].
>
> 2. $\textbf{Choice of models:}$ To recall, ReAct originally showed all results only on the PaLM model (currently decommissioned) and not on any other LLM. Hence, in an effort to understand the working of ReAct while giving it all the possible benefits of doubt, we tested with multiple LLM models, including the most recent OpenAI and Claude models accessible at the time of writing this paper, such as GPT-3.5-Turbo, GPT-3.5-Instruct, GPT-4, GPT-4o, Claude-Opus, and Llama 3.1-8b.
>
> 3. $\textbf{Utility of exemplar-query similarity:}$ In this work, we primarily intended to focus on the supposed usefulness of think tags, i.e., interleaved reasoning traces in multi-step text-based decision making problems. Our hypothesis behind the analysis was to give ReAct the benefit of doubt assuming that it is the reasoning trace (its content and location in the prompt) that leads to increased LLM performance on decision-making domains such as AlfWorld and WebShop. To recall, we note from Table 3 (RQ3) that changing the examples in the prompt alone drops the performance across multiple LLM models in the AlfWorld task, which is completely opposite to the case when we modify the location and content of the think tag in Table 1 (RQ1) and Table 2 (RQ2).  While our results have some relation to other findings regarding the role of examples in the few-shot settings, we wanted to systematically study and show how the reasoning traces, which is the primary claim behind ReAct and all the follow-up works that build on ReAct, is of practically no use and only leads to requiring prompt engineers include these reasoning traces in the examples.
> To conclude, we believe that it will be helpful for practitioners and future works to take these results into account, particularly when designing prompts for text-based decision-making problems, and benefit from avoiding putting any efforts into constructing reasoning traces but rather select the right examples for subsequent problems. We have included this point in our Conclusion and Broader Impact section.
>
>
> [1] Qin, Yujia, et al. "Tool learning with foundation models." ACM Computing Surveys 57.4 (2024): 1-40.
>
> [2] Tong, Peter, et al. "Cambrian-1: A fully open, vision-centric exploration of multimodal llms." Advances in Neural Information Processing Systems 37 (2024): 87310-87356.
>
> [3] Zhao, Andrew, et al. "Expel: Llm agents are experiential learners." Proceedings of the AAAI Conference on Artificial Intelligence. Vol. 38. No. 17. 2024.
>
> [4] Wang, Xingyao, et al. "Executable code actions elicit better llm agents." Forty-first International Conference on Machine Learning. 2024.
>
> [5] ​​Kambhampati, Subbarao, et al. "Position: LLMs can’t plan, but can help planning in LLM-modulo frameworks." Forty-first International Conference on Machine Learning. 2024.
>
> [6] Valmeekam, Karthik, et al. "On the planning abilities of large language models-a critical investigation." Advances in Neural Information Processing Systems 36 (2023): 75993-76005.

---

### Review · Reviewer_WeDX · 2025-04-06

**Summary Of Contributions:**

This submission studies prompting and chain-of-thought strategies in the ReAct (reasoning-action) framework for agentic LLM in simulated planning domains. Through comprehensive evaluation of recent models (GPT-3.5, GPT-4, Claude-Opus, Llama-3.1-8B) on two typical planning tasks (AlfWorld, WebShop), the submission challenges common beliefs that thinking or reasoning might not be the critical factor in some planning scenarios. Instead, the domain consistency of few-shot examples plays a bigger role.

**Audience:**

Yes

**Broader Impact Concerns:**

No concern that requires Broader Impact Statement.

**Claims And Evidence:**

Yes

**Requested Changes:**

- Better organization: It could be better to merge the experiment setup with the results part so that we can introduce the experiment protocols and then the results for each RQ separately. In the current organization, readers have to jump between the two sections to understand each RQ.

- Result reproducibility: For Average Success % numbers, reporting the number of tested instance and the corresponding confidence intervals would be good. Since the tested instances are small and diverse for different settings, we need to include confidence intervals to highlight numbers with potentially high variance and have an apple-to-apple comparison.

- Discussion in Section 5.2: It seems that placing diverse reasoning traces improves the model performance in most cases. Does it imply that models can benefit from more diverse and complicated reasoning traces?

- (Optional) It would be interesting to contact and then report the opinions of ReAct authors for these findings.

**Strengths And Weaknesses:**

Strengths:
- The submission reveals an important and timely finding, questioning the role of reasoning in planning tasks. The studied problem and findings are critical and timely given the rise of agentic LLMs and reasoning LLMs.
- The evaluated prompting strategies are comprehensive, including think token formatting, multi-turn-to-sing-turn reformatting, order reversal, anonymization, example similarity-based alteration, etc. In the LLM planning domain, such a systematic evaluation is needed, and even of more value than papers that propose a single strategy that only works in some selected scenarios.

Weaknesses:
- Evaluation is limited in terms of model selection and task selection. From model selection, the submission can be benefited from evaluation of recent reasoning-enhanced models such as OpenAI o1, o3, Google Gemini Flash Thinking 2.0, and DeepSeek r1. These models are augmented with reasoning processes through <think> tokens in general domains. It would be interesting to see whether they are still limited in generalization in planning scenarios. From the task selection, the submission is heavily based on the study in ReAct (Yao et al, 2022b). It would be interesting to see whether same conclusion holds for tasks beyond those in the original ReAct paper.

- The submission can be improved in terms of better organization, result reproducibility, etc. See Requested Changes below.

---

> ### Author Response · Authors · 2025-04-16
> **Rebuttal**
>
> We would like to thank the reviewer for their thoughtful comments and detailed feedback. We respond to the concerns raised below:
>
> 1. $\textbf{Evaluation with reasoning models:}$ We would like to clarify that the reasoning models such as DeepSeek R1 generate intermediate reasoning tokens themselves such that the model’s performance can improve over a variety of reasoning tasks. However, an important distinction to be made here is that the ReAct framework requires humans in the loop to write the think tag content for the provided in-context examples which can expectedly help the queried LLM to solve the downstream tasks by generating similar think tags.
>
> 2. $\textbf{Better organization:}$ For ease of the reader’s understanding, we have included additional notes to recall the intuition and the designed prompt variations for each RQ in our Results section. We refer the reviewer to Sections 5.1, 5.2, and 5.3 to see the reflected changes.
>
> 3. $\textbf{Result reproducibility:}$ We have included the details on number of instances tested and confidence intervals in the updated PDF in Section 5 (pg 7) and Tables 4,5 and 6 in Appendix Section B.3 (pages 16, 17).
>
> 4. $\textbf{Discussion in Section 5.2:}$ We weren't quite sure what exactly the reviewer had in mind in the question about diversity with reference to Sec 5.2. If the reviewer is referring to the fact that in that section, we experimented with different types of think tags (Failure, Explanation, Placebo, and Ordering), we want to reiterate that our point there is to show that the effects of those tags on the performance seems to be largely independent of the actual tags (or their placement as shown in Section 5.1) even when the English meaning of the think tags is disconnected with what is relevant in the task.

---

### Author Response · Authors · 2025-04-08
**Official Comment**

We thank the reviewers for their comments and thoughtful feedback! We would like to let you know that we are actively working on the responses and revisions. We will get back to you as soon as possible.

---

### Author Response · Authors · 2025-04-16
**Official Comment**

We would like to once again thank all the reviewers for their time and feedback on our work. Along with the responses to the individual comments, we summarize the critical changes here and have also attached an updated PDF with the new changes in $\textcolor{red}{red}$.

1. $\textbf{Detailed analysis of the generated think tags:}$ We have included a new subsection of an example of the ‘think’ tag content for the AlfWorld domain across the different prompt variation settings shown in the work (Section B.4, page 17 onwards).

2. $\textbf{Improved organization:}$ For ease of the reader’s understanding, we have included additional notes to recall the intuition and the designed prompt variations for each research question in our Results section (Sections 5.1, 5.2, and 5.3).

3. $\textbf{Result reproducibility:}$ We have included the confidence intervals for all our experiments across AlfWorld and WebShop domains (Section 5, pg 7 and Tables 4,5 and 6 in Appendix Section B.3, pages 16, 17).

4. $\textbf{Broader Impact statement:}$  We have included a Broader Impact statement at the end of our work discussing the over-reliance on LLMs especially for critical applications, and other considerations that come along with using these models (pg 13).

---

### Decision · Action_Editor_tnLM · 2025-05-11

**Recommendation:** Accept with minor revision

**Comment:**

The work addresses a timely and understudied issue, with carefully designed ablation studies. The conclusions are well-supported and challenge widely held assumptions.

The scope is narrow (limited to ReAct’s original benchmarks), and broader generalization (e.g., newer reasoning-augmented models like O3, R1, Gemini Flash Thinking 2.0) remains unexplored. However, the authors’ focused analysis is sufficient for TMLR’s standards.

The authors have already addressed key reviewer concerns (e.g., adding confidence intervals, clarifying examples, and broader impact statements). Minor revisions should ensure clarity in describing prompt variations and their implications.

**Audience:**

Yes. The findings are highly relevant to TMLR’s audience, particularly researchers working on LLM reasoning, prompt engineering, and sequential decision-making. The paper raises critical questions about the robustness of popular prompting methods and highlights practical implications for designing reliable LLM agents.

**Claims And Evidence:**

Yes. The submission provides systematic evidence through prompt variations and experiments across multiple LLMs (GPT-3.5, GPT-4, Claude-Opus, Llama-3.1-8B) and benchmarks (AlfWorld, WebShop). The core claim-that ReAct-style prompting’s performance stems from exemplar-query similarity rather than inherent reasoning-is supported by:
- Comparable performance when replacing ReAct’s interleaved reasoning traces with placebo or irrelevant content.
- Sharp performance drops when exemplars and queries differ slightly (e.g., synonyms, task variations).
- Minimal impact from altering the order or format of reasoning traces.
While the experiments are limited to domains from the original ReAct work, the methodology is rigorous and directly challenges the foundational claims of ReAct.